# SHUFFLE-R1: EFFICIENT RL FRAMEWORK FOR MULTIMODAL LARGE LANGUAGE MODELS VIA DATA-CENTRIC DYNAMIC SHUFFLE

**Linghao Zhu[1]**  **Yiran Guan[1]**  **Dingkang Liang[1]**  **Jianzhong Ju[2]**  **Zhenbo Luo[2]**
**Bin Qin[2]**  **Jian Luan[2]**  **Yuliang Liu[1]**  **Xiang Bai[1]✉**
[1]Huazhong University of Science and Technology,  [2]MiLM Plus, Xiaomi Inc.
{lh_zhu, yiranguan, dkliang, ylliu, xbai}@hust.edu.cn
{jujianzhong, luozhenbo, qinbin, luanjian}@xiaomi.com
https://xenozlh.github.io/Shuffle-R1

## ABSTRACT

Reinforcement learning (RL) has emerged as an effective post-training paradigm for enhancing the reasoning capabilities of multimodal large language model (MLLM). However, current RL pipelines often suffer from training inefficiencies caused by two underexplored issues: Advantage Collapsing, where most advantages in a batch concentrate near zero, and Rollout Silencing, where the proportion of rollouts contributing non-zero gradients diminishes over time. These issues lead to suboptimal gradient updates and hinder long-term learning efficiency. To address these issues, we propose **Shuffle-R1**, a simple yet principled framework that improves RL fine-tuning efficiency by dynamically restructuring trajectory sampling and batch composition. It introduces (1) Pairwise Trajectory Sampling, which selects high-contrast trajectories with large advantages to improve gradient signal quality, and (2) Advantage-based Batch Shuffle, which increases exposure of valuable rollouts through strategic batch reshuffling. Experiments across multiple reasoning benchmarks show that our framework consistently outperforms strong RL baselines with minimal computational overhead. These results support the potential of data-centric adaptations for more efficient RL training for MLLM.

## 1 INTRODUCTION

Reinforcement learning (RL) has emerged as a powerful tool to enhance large language models (LLMs) to plan, reflect, and generalize, enabling stronger performance in complex reasoning domains such as mathematical problem solving and code generation (Guo et al., 2025; Deepmind, 2025; OpenAI, 2024; Seed, 2025). Notably, DeepSeek-R1 (Guo et al., 2025) leverages reward signals derived exclusively from verifiable outcomes to yield impressive performance gains. Beyond textual tasks, RL has also seen increasing application in various multimodal domains (Li et al., 2025; Liu et al., 2025c;b; Wang et al., 2025c), highlighting its potential to support generalizable reasoning across modalities.

To better incorporate RL in LLM and Multimodal LLM (MLLM), recent studies have proposed various improvements, including better framework optimization (Yu et al., 2025; Wang et al., 2025a; Chu et al., 2025b) and more sophisticated reward optimization (Liu et al., 2025a; Ma et al., 2025). However, most approaches remain confined to static sampling paradigm, where trajectories are sampled and treated uniformly. Such strategy overlooks a crucial insight that not all learning signals are created equal and their informativeness varies and evolves during training. Ignoring variation in signal quality risks the training process being overwhelmed by noisy trajectories and underusing the truly useful signals. An ideal framework should instead concentrate updates on golden signals while filtering out noisy ones. Consequently, it leads us to consider a fundamental question: *Can dynamically prioritizing trajectories provide richer gradient information and lead to more effective training?*

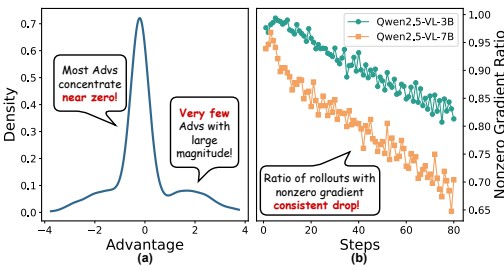
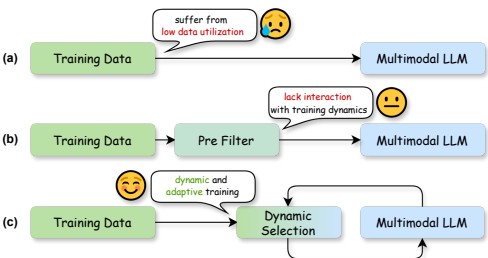

Figure 1: **(a)** Advantage Collapsing, where most advantages concentrate near zero. **(b)** Rollout Silencing, where the ratio of rollouts with non-zero gradient consistently drops.

Figure 2: Pipeline comparison. **(a)** Static paradigm. **(b)** Rule-based pre-filter paradigm. **(c)** Dynamic paradigm can 'interact' with model during training.

Motivated by this question, we investigate into current RL training practices and reveal two critical yet might underexplored limitations. First, **Advantage Collapsing** emerges when most computed advantages cluster excessively near zero, drowning out informative signals from trajectories with large-magnitude advantages, resulting in extremely weak or negligible gradient updates and wiping out decisive ones (Fig. 1(a)). Second, **Rollout Silencing** arises as the fraction of rollouts contributing non-zero gradients steadily declines during training (Fig. 1(b)), leading to catastrophic waste of computation without fully utilizing informative signals. These two findings demonstrate a pressing need for adaptive mechanisms to prioritize, reuse, and reallocate gradient exposure toward informative samples.

In this paper, we present **Shuffle-R1**, a framework that dynamically prioritizes and amplifies critical gradient signals during RL fine-tuning. Guided by the philosophy that *what data the model updates on* is as important *as how it updates*, Shuffle-R1 introduces two effective modules: (1) Pairwise Trajectory Sampling, which selects high-contrast trajectory pairs with large advantages gaps from an extended rollout pool, concentrating on the most discriminative learning signals to mitigate Advantage Collapsing; and (2) Advantage-based Batch Shuffle, which adaptively reshapes training batches to emphasize informative trajectories while down-weighting unhelpful ones, alleviating Rollout Silencing by removing noisy signals and improving computation utilization. Together, our framework embodies the principle of dynamic data prioritization (Fig. 2), enabling adaptive interaction between model and data. Experiments show that Shuffle-R1 substantially improves multimodal reasoning performance, surpassing GPT-4o (Achiam et al., 2024) and Claude-3.7 (Anthropic, 2025) on MathVerse (Zhang et al., 2024b) and MathVista (Lu et al., 2023), and matches GRPO (Shao et al., 2024) with only half the training steps. In summary, our contributions are three-fold:

- We reveal two critical yet underexplored limitations that undermine training efficiency in RL finetuning for MLLM, i.e., Advantage Collapsing and Rollout Silencing.
- We propose Shuffle-R1, a novel and adaptive RL framework that dynamically selects high-contrast trajectories and reshapes training batches to emphasize informative samples.
- Extensive experiments across model scales and both in-domain and out-of-domain benchmarks demonstrate the effectiveness and generalizability of our framework.

These results underscore the importance of rethinking *which data to update on* in RL post-training, moving beyond reward design toward dynamic and adaptive data structuring for more effective reasoning enhancement.

## 2 RELATED WORK

### 2.1 LARGE REASONING MODELS

Researchers have explored various approaches to equip LLM with reasoning ability. Some early studies performed SFT on complex long chain-of-thought data, leading to performance gains on reasoning tasks (Muennighoff et al., 2025; Ye et al., 2025; Guo et al., 2024). However, it has been argued that SFT merely enables the model to memorize the format of reasoning steps and long chains of thought, without fully grasping the ability to reason independently (Chu et al., 2025a; Kang et al.,

2024). Some researchers control the model to generate structured chain-of-thought instead of free generation, achieving systematic step-by-step reasoning output (Xu et al., 2025; Wu et al., 2025; Thawakar et al., 2025) Other works attempted to use test-time scaling like Monte Carlo Tree Search (MCTS) (Yao et al., 2023; Zhang et al., 2024a; Yao et al., 2024; Guan et al., 2026) to facilitate complex reasoning by actively extending the output of the model.

Recently, models such as OpenAI o1/o3 (OpenAI, 2024), DeepSeek-R1 (Guo et al., 2025), Seed-Thinking (Seed, 2025), and Kimi-k1.5 (Kimi, 2025) utilized RL to enable the model to explore independently, stimulating reasoning ability. In particular, DeepSeek-R1-Zero directly conducted RL on pre-trained model without instruction fine-tuning, with verifiable outcome reward functions to replace reward models, achieving remarkable reasoning ability. The training algorithms for RL are also constantly being optimized (Yu et al., 2025; Chu et al., 2025b). Our work focuses on a deeper investigation of the efficiency of RL training and proposes an effective solution to improve both the efficiency and performance of RL training.

## 2.2 REINFORCEMENT LEARNING FOR MLLM

Following the success of DeepSeek-R1, a series of studies have transplanted RL into the training of MLLM and downstream visual tasks, such as Open Vocabulary Object Detection (Liu et al., 2025c), Reasoning Segmentation (Liu et al., 2025b), Video Understanding (Li et al., 2025), Video Localization (Wang et al., 2025c), etc. These works mainly focus on the applicability of RL to downstream tasks. Some other works focusing on improving the general reasoning ability of MLLM have achieved performance improvements on reasoning tasks by collecting a large amount of high-quality data (Meng et al., 2025; Huang et al., 2025; Yang et al., 2025; Zhang et al., 2025; Peng et al., 2025; Tan et al., 2025). These works mainly focus on the organization of high-quality reasoning data and the balance between SFT and RL in the training process.

Some researchers who conduct in-depth research on the RL mechanism have optimized the RL training process from various aspects, including adding contrastive reward mechanism (Li et al., 2025), actively introducing reflection tokens during rollouts (Wang et al., 2025a), optimizing the RL objective function and gradient update mechanisms (Chu et al., 2025b), and introducing more diverse rollouts (Liu et al., 2025a; Yao et al., 2025). The core objective of these works is to optimize the RL training process. In our work, we propose a novel training framework that introduces dynamic and adaptive selection and resampling of queries and rollouts, reshaping the data distribution for better training efficiency and model performance.

## 3 METHOD

In this section, we begin by further analyzing the existing drawbacks in training. Then, as illustrated in Fig. 4, we introduce **Shuffle-R1**, which optimizes the training process through two crucial modules: (1) Pairwise Trajectory Sampling and (2) Advantage-based Batch Shuffle.

### 3.1 PRELIMINARIES

Policy gradient algorithm is a widely used RL method for MLLM, aiming to maximize the expectation of return from the environment. For a given query $q$, $N$ independent responses $O = \{o_1, o_2, ..., o_N\}$ are sampled from the old policy model $o \sim \pi_{\theta'}(q)$. Each response receives corresponding reward $R = \{r_1, r_2, ..., r_N\}$ computed via verifiable reward functions. Advantages $\hat{A} = \{\hat{A}_1, \hat{A}_2, ..., \hat{A}_N\}$ are then estimated to guide policy updates. The core objective is defined as:

$$\mathcal{J}(\theta) = \mathbb{E}_{q \sim \mathcal{D}, \{o_i\}_{i=1}^{N} \sim \pi_{\theta'}(\cdot|q)} \frac{1}{\sum_{i=1}^{N} |o_i|} \sum_{i=1}^{N} \sum_{t=1}^{|o_i|} \left\{ \min \left[ \gamma_t(\theta) \hat{A}_i, \text{clip}\left(\gamma_t(\theta), 1 - \epsilon, 1 + \epsilon\right) \hat{A}_i \right] \right\}, \quad (1)$$

where

$$\gamma_t(\theta) = \frac{\pi_\theta(o_{i,t}|q, o_{i,<t})}{\pi_{\theta'}(o_{i,t}|q, o_{i,<t})}, \qquad \hat{A}_i = \frac{r_i - \text{mean}(R)}{\text{std}(R)} \qquad (2)$$

and $\epsilon$ is a clipping hyperparameter to prevent training collapse.

Despite its practicality, this static RL paradigm has notable drawbacks: advantages often concentrate near zero, yielding weak gradient signals, and the fraction of rollouts with non-trivial updates diminishes as training progresses. These issues highlight the need for a more flexible, dynamic and efficient training framework.

## 3.2 PROBLEM ANALYSIS

**Advantage Collapsing.** Our probe analysis reveals that, contrary to the ideal scenario, most rollouts exhibit advantages sharply concentrated around zero in standard RL paradigm, leading to the Advantage Collapsing phenomenon. This concentrated distribution weakens gradient signals, as only a few rollouts with high-magnitude advantages drive meaningful updates. While simply increasing the number of rollouts can partially mitigate this issue by increasing the chance of sampling valuable trajectories (Fig. 3(a)), it substantially increases computational overhead without addressing the root cause. These findings highlight the need for a dynamic mechanism that adaptively selects valuable rollouts to improve training efficiency.

**Rollout Silencing.** We further observe a notable Rollout Silencing phenomenon, where the fraction of rollouts contributing non-zero gradients notably declines during training. It arises from factors such as zero advantages, gradient clipping, and excessive truncation, exposing the limits of static sampling paradigm. Tracking accuracy of queries in different difficulty reveals that simple queries converge early while difficult ones remain inaccurate (Fig. 3(b)), both failing to generate informative rollouts. Moreover, standard pipelines use each rollout only once, preventing full exploitation of valuable data. Consequently, it is crucial to design a dynamic strategy that discards ineffective rollouts while reusing informative ones.

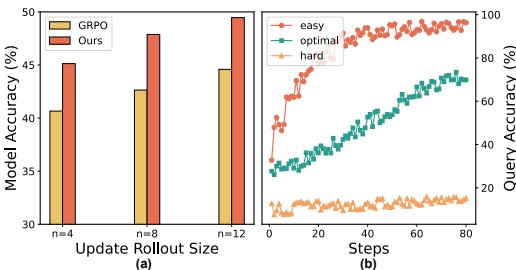

Figure 3: **(a)** Model accuracy improves with larger rollout sizes. **(b)** Queries with different difficulties have varying accuracy during training, yielding rollouts with different diversity and qualities.

## 3.3 PAIRWISE TRAJECTORY SAMPLING

To mitigate Advantage Collapsing, we seek to select trajectories that offer stronger learning signals. Inspired by the observation that a larger rollout pool increases the probability of capturing high-advantage samples, we propose Pairwise Trajectory Sampling (PTS), a data-centric module to selectively amplify valuable learning signals. Rather than evaluating trajectories in isolation, PTS organizes candidate rollouts into structured contrastive pairs. This pairing mechanism captures both high and low advantage signals jointly, forming informative "positive-negative" pairs. Only pairs with the largest advantage contrast are then retained for training. This process ensures that limited update bandwidth is focused on trajectories that are both diverse and gradient-rich.

Given a query $q$ and a rollout size of $2N$, the rollout trajectories group is denoted as $O = \{o_i\}_{i=1}^{2N}$. The corresponding reward and advantage sets are $R = \{r_i\}_{i=1}^{2N}$ and $A = \{\hat{A}_i\}_{i=1}^{2N}$, respectively. To identify informative trajectory pairs, our proposed pairing mechanism follows a straightforward 'max-min' pairing principle by matching the trajectory with the highest advantage to that with the lowest, the second highest to the second lowest, and so on. We denote the sorted advantage values in descending order as:

$$A_s = \{\hat{A}_{(i)}\}_{i=1}^{2N}, \quad \text{where } \hat{A}_{(1)} \geq \hat{A}_{(2)} \geq \cdots \geq \hat{A}_{(2N)}. \tag{3}$$

Based on this ordering, we construct the pairing set as:

$$P = \{(o_{(i)}, o_{(2N-i+1)})\}_{i=1}^N. \tag{4}$$

In this scheme, the original $2N$ rollouts are easily sorted and reorganized into $N$ pairs. The top-ranked pairs typically consist of trajectories with high-magnitude but opposite-sign advantages, forming contrastive pairs akin to 'positive-negative' samples. In contrast, the bottom-ranked pairs involve trajectories with advantages closer to zero.

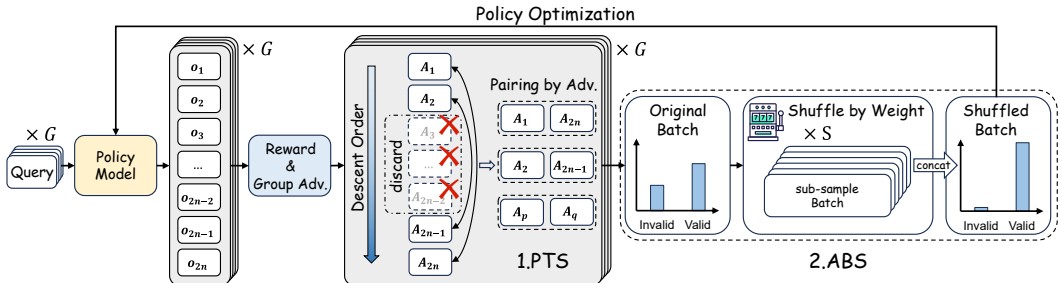

Figure 4: Overview of our proposed Shuffle-R1. After advantage calculation, we first conduct Pairwise Trajectory Sampling to obtain valuable trajectory pairs from original rollout pool, then perform Advantage-based Batch Shuffle to reshape the distribution of valid trajectories in a batch.

As implied by Eq. 1, trajectories with higher absolute advantages contribute more significantly to the gradient update, while those with near-zero advantages have negligible impact. We apply a simple top-$k$ sampling strategy to select a subset of valid pairs:

$$P_v = \{(o_{(i)}, o_{(2N-i+1)})\}_{i=1}^M, \quad M = \alpha N, \ \alpha \in (0,1). \tag{5}$$

where $\alpha$ is a hyperparameter controlling the sampling ratio from the pairing set. Only the trajectories within this valid set $P_v$ are used for the subsequent gradient update.

By introducing a structured contrastive sampling scheme, PTS enables more effective trajectory selection from a broader exploration space without increasing the gradient computation cost. The contrastive structure not only filters out low-signal trajectories, but also sharpen the model's policy gradient through direct comparison. PTS shifts the focus of RL fine-tuning from uniform exploration to gradient-informed selection, representing a principled step toward more efficient data usage in RL training.

### 3.4 ADVANTAGE-BASED BATCH SHUFFLE

While PTS mitigates Advantage Collapsing and improve RL training performance, the Rollout Silencing issue remains unresolved. To overcome this issue, we propose Advantage-based Batch Shuffle (ABS) module that dynamically reshapes training batches to prioritize and reinforce high-value samples. Rather than relying on static data flow, ABS adaptively redistributes trajectories within each training batch, enabling more frequent updates to trajectories with high learning utility. Built on top of PTS, it serves to magnify the gradient exposure of informative samples, reshaping the training data distribution to achieve better data utilization and training efficiency.

Denote a data batch provided by PTS:

$$B = \{p_i^g : (o_{i,1}^g, \hat{A}_{i,1}^g, o_{i,2}^g, \hat{A}_{i,2}^g, q^g)\}_{i=1\sim M, g=1\sim G}, \tag{6}$$

with batch size of $M \times G$, In the standard gradient update process, $B$ is sequentially divided into $K$ mini-batches, each contains $MG/K$ samples.

In our ABS module, we first assign an importance weight to each pair $p_j \in B$ based on the sum of the absolute advantages:

$$W(p_j) = |\hat{A}_{j,1}| + |\hat{A}_{j,2}|. \tag{7}$$

These weights are then normalized to form a sampling distribution $\Phi$ over the entire batch $B$:

$$\Phi(p_j) = \frac{W(p_j)}{\sum_{k=1}^{|B|} W(p_k)}. \tag{8}$$

Based on the sampling distribution, we perform $S$ sub-sampling from original batch $B$, each sub-sampling has a capacity of $T$ pairs ($2T$ trajectories):

$$B_s = \{p_{s,t}\}_{t=1}^T, \quad \text{s.t. } p_{s,t} \neq p_{s,t'}, \forall t \neq t'. \tag{9}$$

All the sub-sampling batches are sequentially combined to form the *reshuffled* batch $B' = \bigcup_{s=1}^S B_s$. During the ABS process, we set $|B'| = |B|$, i.e., $S \times T = MG$ to ensure the reshuffled batch

Table 1: Performance of Shuffle-R1 trained on Geometry3K dataset.

| Method | Geo3K | Math Avg. | HallBench | ChartQA |
|---|---|---|---|---|
| Qwen-3B | 25.79 | 41.71 | 59.83 | 73.08 |
| + GRPO | 42.64 | 46.74 | 63.09 | 76.20 |
| + DAPO | 45.09 | 48.08 | 63.24 | 76.70 |
| + GSPO | 43.22 | 47.26 | **63.67** | 75.12 |
| **+ Ours** | **47.88**(+22.09) | **48.70**(+6.99) | 63.19(+3.36) | **77.04**(+3.06) |
| Qwen-7B | 38.12 | 49.82 | 65.19 | 79.84 |
| + GRPO | 52.60 | 53.13 | 68.56 | 80.84 |
| + DAPO | 54.43 | 54.19 | 69.29 | 81.20 |
| + GSPO | 52.83 | 54.27 | 69.48 | 80.96 |
| **+ Ours** | **55.89**(+17.77) | **54.63**(+4.81) | **69.51**(+4.32) | **81.64**(+1.80) |

Table 2: Performance of Shuffle-R1 trained on K12 dataset.

| Method | K12 | Math Avg. | HallBench | ChartQA |
|---|---|---|---|---|
| Qwen-3B | 42.42 | 41.71 | 59.83 | 73.08 |
| + GRPO | 59.19 | 48.71 | 64.14 | 77.12 |
| + DAPO | 61.42 | 49.75 | 65.08 | 77.00 |
| + GSPO | 60.44 | 48.76 | 64.14 | 77.56 |
| **+ Ours** | **62.22**(+19.80) | **50.05**(+8.34) | **65.72**(+5.89) | **78.28**(+5.20) |
| Qwen-7B | 52.13 | 49.82 | 65.19 | 79.84 |
| + GRPO | 66.15 | 54.47 | 67.75 | 82.48 |
| + DAPO | 68.35 | 54.52 | 68.66 | 82.52 |
| + GSPO | 67.44 | 54.77 | 69.13 | 82.04 |
| **+ Ours** | **68.78**(+16.65) | **55.02**(+5.20) | **69.87**(+4.68) | **82.60**(+2.76) |

matches the same size as the original batch. The reshuffled batch will maintain the gradient update paradigm of the original method.

The ABS module optimizes the learning process through Advantage-aware Shuffling and Sub-batch Resampling. It increases the update frequency of trajectories with higher advantages, maintains diversity while reinforcing high-value samples through repeated exposure. Together, these designs transform each batch into a soft-prioritized structure that better reflects training signal utility.

# 4 EXPERIMENTS

## 4.1 EXPERIMENTAL SETUP

**Datasets and Benchmarks.** We first conduct our experiments on Geometry3K dataset (Lu et al., 2021) ($2.1k$ training samples, 'Geo3K' for short) and a subset of MMK12 dataset (Meng et al., 2025) containing the same amount of data ('K12' for short), to investigate model performance on limited training resources. To assess the scalability and effectiveness on a larger corpus, we then conduct experiments with MM-Eureka dataset (Meng et al., 2025). Specifically, we construct a $30k$-sample training set by combining the full Geo3K dataset with $27k$ randomly selected samples from MM-Eureka. All training samples are in free-form format.

We first perform evaluation on in-domain test set of Geometry3K and MMK12. Further, as RL is famous for its strong generalizability, we evaluate our model's performance on the following representative visual reasoning benchmarks: MathVerse (Zhang et al., 2024b), MathVision (Wang et al., 2024), WeMath (Qiao et al., 2025), MathVista (Lu et al., 2023), HallucinationBench (Guan et al., 2024) and ChartQA (Masry et al., 2022). These benchmarks span across math reasoning, visual perception, and chart understanding. We use MathRuler to evaluate questions with free-form ground truths and Gemini-2.0-Flash-001 (Deepmind, 2025) to evaluate questions with multi-choice ground truths. More information in Appendix E.

**Implementation Details.** We use EasyR1 (Yaowei et al., 2025) as our training codebase. We employ Qwen2.5-VL-3B/7B-Instruct (Bai et al., 2025) as base model to verify our method's generalizability on model scales. Parameters of vision encoder are kept frozen. We set update batch size to 128 and rollout batch size ($G$) to 512. Rollout temperature is set to 1.0 and learning rate is set to $1e-6$. All experiments are conducted on $8\times$ 80G GPUs. For each query, we generate $2N=16$ rollouts. For PTS, we construct $N=8$ pairs and select the top $M=4$ pairs, corresponding to a retention ratio of $\alpha = 0.5$, striking a balance between training cost and exploration space. For ABS, we set $T=256$ pairs (512 query-response trajectories) for each sub-sampling batch. We construct shuffled batch by performing $S=8$ rounds of shuffle. Decoding temperature for evaluation is set to 0.5, and we report average pass@1 accuracy of 8 tests to reduce randomness.

## 4.2 MAIN RESULTS

**Comparison with Representative Algorithms.** We compare our method with GRPO (Shao et al., 2024) and DAPO (Yu et al., 2025). On Geometry3K (Tab. 1), our 3B model reaches 47.88% accuracy, outperforming GRPO by 5.2% and DAPO by 2.7%; the 7B model achieves 55.89%, with

Table 3: Model performance on representative visual reasoning benchmarks. Models marked with '*' are evaluated using our own evaluation scripts with vLLM. †Vision-R1-7B used WeMath and MathVision as training data, its performance on these benchmarks are omitted. Best performance of RL-only models marked with **Bold**, second best with underline.

| Model | MathVerse | MathVision | MathVista | WeMath | HallBench | ChartQA | Avg. |
|---|---|---|---|---|---|---|---|
| *Close-source* | | | | | | | |
| GPT-4o (Achiam et al., 2024) | 50.8 | 30.4 | 63.8 | 68.8 | 55.0 | - | - |
| o1 (OpenAI, 2024) | 57.0 | 60.3 | 73.9 | - | - | - | - |
| Gemini-2.0 pro (Deepmind, 2025) | 67.3 | 48.1 | 71.3 | - | 49.8 | - | - |
| Claude-3.7-Sonnet (Anthropic, 2025) | 52.0 | 41.3 | 66.8 | 72.6 | 55.4 | - | - |
| *Open-Source SFT* | | | | | | | |
| InternVL-2.5-8B (Chen et al., 2025b) | 39.5 | 17.0 | 64.5 | - | 50.1 | 79.1 | - |
| InternVL-3-8B (Zhu et al., 2025) | - | 29.3 | 71.6 | - | 49.9 | 86.6 | - |
| Qwen2.5-VL-3B* (Bai et al., 2025) | 34.8 | 21.9 | 58.4 | 51.7 | 59.8 | 73.1 | 49.9 |
| Qwen2.5-VL-7B* (Bai et al., 2025) | 42.6 | 25.8 | 67.4 | 63.5 | 65.2 | 79.8 | 57.4 |
| *Cold-Start + RL* | | | | | | | |
| R1-VL-7B* (Zhang et al., 2025) | 40.1 | 24.3 | 62.3 | 59.8 | 60.9 | 76.1 | 53.9 |
| Vision-R1-7B*† (Huang et al., 2025) | 46.1 | - | 70.8 | - | 57.8 | 83.1 | - |
| R1-OneVision-7B* (Yang et al., 2025) | 43.0 | 24.8 | 61.2 | 60.6 | 66.4 | 77.8 | 55.2 |
| OpenVLThinker-7B* (Deng et al., 2025) | 46.4 | 24.8 | 69.7 | 67.2 | 59.1 | 78.4 | 57.6 |
| VLAA-Thinker-7B* (Chen et al., 2025a) | 48.9 | 26.3 | 69.9 | 67.7 | 67.5 | 80.1 | 60.1 |
| *Zero RL* | | | | | | | |
| MM-Eureka-Qwen-7B* (Meng et al., 2025) | 49.6 | 27.4 | 70.6 | 67.4 | 66.7 | 79.0 | 60.1 |
| MMR1-Math-7B* (Leng et al., 2025) | 39.2 | **31.9** | 71.5 | 70.7 | 69.6 | 82.0 | 60.8 |
| ThinkLite-VL-7B* (Wang et al., 2025b) | 45.2 | 28.0 | 72.4 | 69.3 | 70.2 | 82.0 | 61.2 |
| VL-Rethinker-7B* (Wang et al., 2025a) | 51.7 | 29.7 | 72.0 | 70.1 | 69.9 | 79.0 | 62.1 |
| NoisyRollout-7B-K12* (Liu et al., 2025a) | 50.1 | 28.0 | 70.9 | 70.8 | 70.1 | 81.4 | 62.1 |
| **Shuffle-R1-Qwen-3B (Ours)** | 44.2 | 26.8 | 70.4 | 66.5 | 69.2 | 79.9 | 59.5 |
| **Shuffle-R1-Qwen-7B (Ours)** | **53.9** | 30.0 | **77.0** | **72.3** | **71.0** | **84.1** | **64.7** |

gains of 3.3% and 1.4%, respectively. On out-of-domain math reasoning tasks, our method further improves average accuracy by 1.96% (3B) and 1.5% (7B) over GRPO, also surpassing DAPO. Consistent gains are observed on HallusionBench and ChartQA, where data distribution diverges more from training. Similar trends hold on the K12 experiments (Tab. 2), with improvements in both in-domain and out-of-domain settings, underscoring the robustness of our framework across tasks and distributions. Compared with the latest GSPO (Zheng et al., 2025), which replaces token-level importance sampling with sequence-level importance sampling, our method also demonstrates superior performance on both in-domain and out-of-domain tasks. All the performance results above indicate our framework's generalizability on different data distributions and model scales, highlighting the effectiveness of dynamic RL training paradigm. More results in Appendix C.

**Comparison with RL-based models.** We conduct larger scale experiments on MM-Eureka dataset. As shown in Tab. 3, trained with $30k$ selected data from diverse sources for 150 steps, our 7B model exhibits a substantial accuracy gain over the base model (Qwen2.5-VL-7B). Moreover, it outperforms a series of open-source 7B competitors who also adopt RL training strategies, e.g. MM-Eureka with direct RL and VLAA-Thinker with RL after cold-start SFT. Notably, our model achieves competitive or superior performance on several benchmarks compared to leading close-source models, for

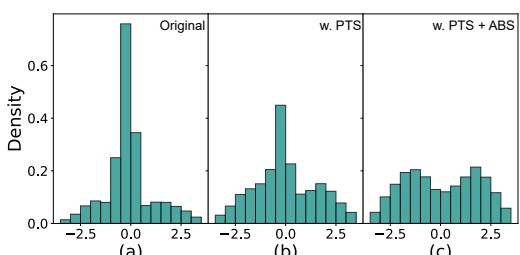

Figure 5: Advantage distribution in a training batch of GRPO and our framework.

instance, Claude-3.7-Sonnet (Anthropic, 2025) and GPT-4o (Achiam et al., 2024). Under the same setting, our 3B variant also demonstrates strong performance, even outperforming several 7B models on certain benchmarks. These results highlight the superiority of our proposed approach in boosting the training efficiency in reinforcement learning.

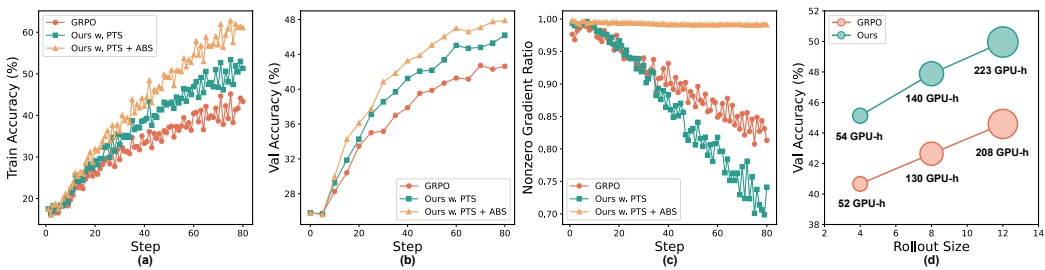

Figure 6: **(a)** Training accuracy of Shuffle-R1 on Geo3K. **(b)** Validation accuracy of Shuffle-R1 on Geo3K. **(c)** Token utilization rate of Shuffle-R1 on Geo3K. **(d)** Shuffle-R1 achieves better performance with minimal extra time cost.

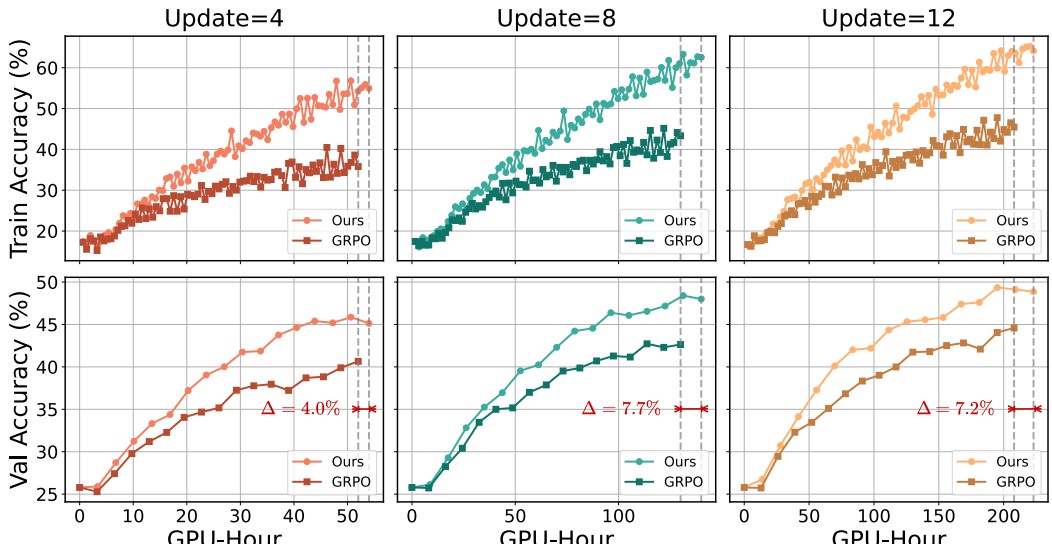

Figure 7: Wall-Clock Training Curve of Shuffle-R1 compared with GRPO.

**Efficiency Analysis.** The improvement in model performance mainly stems from better training efficiency. Advantage distribution analysis in Fig. 5 confirms that, PTS effectively mitigates Advantage Collapsing by increasing the proportion of large-magnitude advantages. ABS further optimizes the batch composition, enabling the model to focus on more informative trajectories.

Fig. 6(a) and (b) further probe into training dynamics and demonstrate that our framework consistently achieves higher training and validation accuracy, reaching comparable performance as GRPO with as little as half the training steps. Moreover, our framework effectively mitigates the issue of "Rollout Silencing" shown in Fig. 6(c), maintaining a high token utilization rate across all training stages. Fig. 6(d) further illustrates the favorable trade-off between training scale and computational cost of our approach, uplifting training performance by a large margin.

Fig. 7 illustrates the actual wall-clock GPU time between GRPO and Shuffle-R1. Under the same update size and total training steps, Shuffle-R1 achieves substantially higher train/val accuracy than GRPO in the early training stage. More importantly, the total GPU time of Shuffle-R1 only increases by $4\% \sim 7.7\%$ relative to GRPO. When targeting the same accuracy as GRPO, Shuffle-R1 requires roughly half the number of training steps and approximately 60% of the total wall-clock time.

### 4.3 ABLATION STUDY

We conduct ablation experiments on Qwen2.5-VL-3B-Instruct using Geometry3K, focusing on two objectives: (1) assessing the contribution of each component in our framework, and (2) validating the impact of key designs. More detailed results are reported in Appendix C.

**Component-wise Contribution.** We evaluate the effectiveness of PTS and ABS by incrementally adding them to the baseline. As shown in Tab. 4, On in-domain Geometry3K test set, PTS lifts accuracy from 42.64% to 46.21% (+3.57). Adding ABS yields a further +1.67, reaching 47.88%. Similar improvement trends appear on out-of-domain benchmarks: on the math reasoning set, the full setting (PTS + ABS) attains 48.70% vs. 46.74% with GRPO (41.71% before RL training); on ChartQA, we also observe effective performance gain (77.04% vs. 76.20%, with 73.08% before RL training) despite larger distribution shift. Our method also demonstrates improved performance in HallusionBench. To isolate the contribution of ABS, we conducted an experiment where ABS is applied to trajectory pairs formed by uniform random sampling instead of PTS. This setting (Tab. 4 line 4) also demonstrates notable improvement over the baseline, confirming its effectiveness.

**Analysis of Pairwise Trajectory Sampling.** One core mechanism of the PTS lies in the structured contrastive sampling scheme. To validate the effectiveness of our bidirectional, contrastive sampling scheme, we compare PTS against three alternative strategies: (1) One-way Positive Sampling (only select trajectories with highest advantage); (2) One-way Negative Sampling (only select trajectories with lowest advantage); and (3) Unbiased Random Sampling. All settings maintained a consistent sampling ratio (8 valid trajectories from 16 rollouts) to ensure fairness. We disable ABS since it relies on the pairing result of PTS. As shown in Tab. 5, model trained with PTS receives a consistent performance gain on both in-domain and out-of-domain tasks, while both one-way positive and negative sampling result in a performance decline even below the GRPO baseline, demonstrating the effectiveness and rationality of our design. Unbiased random sampling only receives minor improvement over baseline, far behind the effectiveness of PTS.

Table 4: Ablation on effectiveness of PTS and ABS.

| GRPO | PTS | ABS | Geo3k | Math Avg. | HallBench | ChartQA |
|------|-----|-----|-------|-----------|-----------|---------|
|      |     |     | 25.79 | 41.71     | 59.83     | 73.08   |
| ✓    |     |     | 42.64 | 46.74     | 63.09     | 76.20   |
| ✓    | ✓   |     | 46.21 | 47.64     | **63.40** | 76.52   |
| ✓    |     | ✓   | 46.82 | 47.79     | 62.77     | 75.12   |
| ✓    | ✓   | ✓   | **47.88** | **48.70** | 63.19 | **77.04** |

Table 5: Ablation on rationality of PTS and ABS.

| Setting | Geo3k | Math Avg. | HallBench | ChartQA |
|---------|-------|-----------|-----------|---------|
| Qwen2.5-VL-3B | 25.79 | 41.71 | 59.83 | 73.08 |
| + GRPO | 42.64 | 46.74 | 63.09 | 76.20 |
| *Ablation on PTS* | | | | |
| + only max | 41.26 | 44.77 | 63.30 | 75.64 |
| + only min | 23.36 | 41.52 | 60.98 | 74.36 |
| + random pick | 43.53 | 46.62 | 63.19 | 76.00 |
| **+ PTS** | **46.21** | **47.64** | **63.40** | **76.52** |
| *Ablation on ABS* | | | | |
| + random shuffle | 46.05 | 47.40 | **63.19** | 76.60 |
| + reorder | 46.28 | 47.64 | 63.09 | 76.64 |
| **+ ABS** | **47.88** | **48.80** | 63.19 | **77.04** |

**Analysis of Advantage-based Batch Shuffle.** ABS introduces Advantage-aware shuffling and Inter-sub-batch resampling to reshape the training batch. To validate their effectiveness, we designed two contrastive experiments: (1) Unbiased Shuffle: using uniformly distributed sampling weights to perform shuffle strategy. and (2) Static Reorder: randomly reorder the training batch without sub-sampling, maintaining the original data distribution. We enable PTS during training as ABS relies on its pairing result. As shown in Tab. 5, model trained with ABS significantly outperforms the contrastive settings. The unbiased shuffle setting even performs worse compared to the PTS-only setting, demonstrating significance of advantage-weighting. The static reorder setting has no improvement compared to PTS-only setting, as they have the same data distribution.

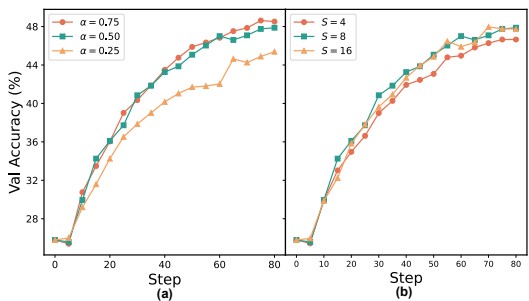

Figure 8: Ablation on key hyper parameters. **(a)** Effect of different sampling ratio $\alpha$. **(b)** Effect of different shuffle times $S$.

**Hyperparameters.** We investigate the impact of two key hyperparameters in our framework: (1) the sampling ratio ($\alpha$) in PTS, and (2) the shuffle times ($S$) in ABS. For $\alpha$, we fix $S = 8$ and test values of 0.25, 0.5, and 0.75 (i.e., selecting 4, 8, and 12 samples from 16 rollouts). As shown in

Table 6: Extension experiments on Qwen2.5-VL-32B.

| Setting | MathVerse | MathVision | MathVista | WeMath | HallBench | ChartQA | Avg. |
|---|---|---|---|---|---|---|---|
| Qwen2.5-32B | 57.0 | 38.2 | 75.4 | 72.9 | 71.3 | 80.7 | 65.9 |
| + GRPO | 58.4 | 39.3 | 77.3 | 75.9 | 70.3 | 83.8 | 67.4 |
| + Ours | **59.0** | **41.2** | **79.5** | **77.9** | **72.2** | **84.9** | **69.1** |

Table 7: Extension experiments on Referring Expression Comprehension task.

| Setting | RefCOCO(testA) | RefCOCO(testB) | RefCOCO+(testA) | RefCOCO+(testB) | RefCOCOg(test) |
|---|---|---|---|---|---|
| Qwen2.5-3B | 86.09 | 75.64 | 81.71 | 66.93 | 72.39 |
| +GRPO | 89.90 | 81.33 | 85.94 | 70.97 | 81.45 |
| +Ours | **91.83** | **84.31** | **87.84** | **76.27** | **86.07** |

Fig. 8(a), both $\alpha = 0.75$ and $\alpha = 0.5$ yield strong performance, while $\alpha = 0.25$ lags behind. We attribute this to over-pruning, where filtering out rollouts too aggressively may reduce data diversity. We choose $\alpha = 0.5$ for a balance between signal quality and computational efficiency. For $S$, we fix $\alpha = 0.5$ and vary the shuffle times as 4, 8, and 16. Fig. 8(b) shows that performance improves with increasing $S$, but saturates beyond $S = 8$. This suggests that moderate resampling enhances data exposure, but too many shuffles may offer diminishing returns. More details in Appendix C.

## 4.4 EXTENSION EXPERIMENTAL RESULTS

**Scaling to 32B Model.** We conduct experiments on Qwen2.5-VL-32B to inveritage the scalability and generalizability of Shuffle-R1 on larger models. We train the 32B variant on the selected $30k$ data for 50 steps (to save training resources cost). Results in Tab. 6 demonstrate that Shuffle-R1 also performs well on 32B scale.

**Referring Expression Comprehension.** For tasks beyond math/visual reasoning, we conduct an experiment on Referring Expression Comprehension (REC). We train Qwen2.5-VL-3B on 60K data randomly selected from RefCOCO/RefCOCOg/RefCOCO+ using a IoU-based soft reward. Results in Tab. 7 show that Shuffle-R1 can be well adapted to soft-reward tasks and beyond.

**Language-only Reasoning.** While our initial analysis focused on the RL training dynamics of MLLMs, we conduct an early extension experiment to validate the feasibility of Shuffle-R1 on LLMs. We train Qwen2.5-Math-1.5B/7B-Base (Yang et al., 2024) on Open-S1 dataset (Dang & Ngo, 2025) for 150 steps and evaluate model performance on Math12K (Hendrycks et al., 2021), AIME24, MATH500 (Lightman et al.,

Table 8: Extension experiments on LLM.

| Setting | Math12K | AIME24 | MATH500 | GSM8K | GPQA | Olymp. |
|---|---|---|---|---|---|---|
| | *Qwen2.5-Math-1.5B-Base* | | | | | |
| + GRPO | 67.6 | 10.0 | 66.0 | 74.0 | 25.7 | 33.3 |
| + Ours | 70.4 | 16.6 | 71.0 | 79.6 | 30.8 | 36.8 |
| | *Qwen2.5-Math-7B-Base* | | | | | |
| + GRPO | 74.6 | 20.0 | 76.4 | 84.8 | 36.3 | 39.7 |
| + Ours | 78.2 | 23.3 | 79.4 | 89.5 | 37.3 | 41.4 |

2023), GSM8K (Cobbe et al., 2021), GPQA-Diamond (Rein et al., 2024) and OlympiadBench (He et al., 2024). As shown in Tab. 8, our framework delivers significant improvements on all the benchmarks compared to GRPO, demonstrating its potential effectiveness on text-only LLMs.

## 5 CONCLUSION

In this paper, we introduce Shuffle-R1, a simple but effective framework designed to improve the training efficiency of reinforcement learning of multimodal large language models. Through Pairwise Trajectory Sampling and Advantage-based Batch Shuffle, our framework significantly outperforms representative algorithms and models in both in-domain and out-of-domain tasks, demonstrating the value of data-centric adaptive design. We hope that our motivations, method, and findings are helpful for further research.

ACKNOWLEDGMENT

This work was done during the research internship of Linghao Zhu, Yiran Guan, and Dingkang Liang at Xiaomi.

This work was supported by the National Natural Science Foundation of China (NSFC) under Grants No. 62225603, 62576147 and 623B2038.

ETHICS STATEMENT

All training and evaluation data used in this project are collected from public academic datasets. All the pre-trained model checkpoints and code framework employed are forked from open-source research projects. During the research process, we strictly adhered to the open-source licenses and usage requirements of all relevant parties. No non-public / private data was used.

Furthermore, all authors strictly complied with the ICLR Code of Ethics throughout the research process and affirm that no activities violating these ethical standards were involved.

REPRODUCIBILITY STATEMENT

We have reported the base model checkpoints, code framework, training data, evaluation benchmarks and metrics, and core hyperparameter settings used in the project in the main paper. A detailed pseudo code explaining our framework flow is provided in Appendix A. More detailed hyperparameter settings and information about the training infrastructure are provided in Appendix D. In addition, we have also included the prompts used during training and inference, examples of the training and evaluation data, as well as examples of the model inference results in Appendix E, F and G.

LLM USAGE STATEMENT

During this research process, the usage of LLM is strictly restricted and was used in limited ways: (1) GitHub Copilot was employed during code development for code checking and bug fixes; (2) LLM-based tools were used during paper writing for language refinement and grammar correction, and (3) LLM judges were applied as a part of evaluation metrics, including key answer extraction and correctness assessment.

All core components of this work, including research motivation, algorithm design, core code implementation, data collection, and experiment analysis, and experimental conclusions, were conducted entirely and independently by the authors without reliance on LLMs.

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

## A  PSEUDO CODE

Here, we provide the pseudo code of Shuffle-R1 for readers to better understand the pipeline flow, and for better transparency in algorithm understanding and reproducibility.

---

**Algorithm 1** Shuffle-R1 Workflow

---

1: **Input:** $\mathcal{Q}$ (queries), $\pi_\theta$ (policy), $2N$ (rollouts per query), $\alpha$ (sampling ratio), $S$ (shuffle rounds)
2: **Output:** Optimized batch $\mathcal{B}'$ for gradient update
3: Initialize global batch $\mathcal{B} \leftarrow \emptyset$
4: **for** each $q \in \mathcal{Q}$ **do**                                                                                                          ▷ **PTS Phase**
5:       Generate $\{o_i\}_{i=1}^{2N} \sim \pi_\theta(q)$
6:       Compute $\{\hat{A}_i\}$ via $R = RewardFunc(\{o_i\}, q)$
7:       Sort pairs: $\{(o_{(i)}, o_{(2N-i+1)})\}_{i=1}^N \leftarrow \text{MaxMinPair}(\{\hat{A}_i\})$
8:       Retain top-$\lfloor \alpha N \rfloor$ pairs: $\mathcal{B}_q \leftarrow \{(o_{(k)}, o_{(2N-k+1)})\}_{k=1}^{\lfloor \alpha N \rfloor}$
9:       Aggregate: $\mathcal{B} \leftarrow \mathcal{B} \cup \mathcal{B}_q$
10: **end for**
11: Compute $W_j = |\hat{A}_p| + |\hat{A}_q| \; \forall (o_p, o_q) \in \mathcal{B}$                                                        ▷ **ABS Phase**
12: Calculate $P_j = W_j / \sum W_k$ for weighted sampling
13: $\mathcal{B}' \leftarrow \text{ShuffleSample}(\mathcal{B}, P_j, S)$ with $S \times T = |\mathcal{B}|$
14: **return** $\mathcal{B}'$                                                                                                        ▷ **Jointly optimized training data**

---

## B  PROMPT DESIGN

We use a "Thinking prompt" to explicitly control the output format of the model, which requires the model to output its thinking process within special tokens `<think>` and `</think>`, and mark the final answer with `\boxed{}`. In practice, we keep the system prompt of Qwen2.5-VL (Bai et al., 2025), and insert the "Thinking prompt" at the beginning of user message. We keep the training and evaluation prompt in the same format. The full structure of instruction prompt is as follows:

> **Prompt Example**
>
> **SYSTEM**:
> You are a helpful assistant.
> **USER**:
> You FIRST think about the reasoning process as an internal monologue and then provide the final answer. The reasoning process MUST BE enclosed within `<think>` `</think>` tags. The final answer MUST BE put in `\boxed{}`. **<QUESTION>**

## C  MORE EXPERIMENTAL RESULTS

**Detailed Model Performance.**    We provide a more detailed performance of models trained on Geometry3K and K12 dataset in Tab. 9, reporting model performance on each out-of-domain benchmarks as a supplement to "Math Avg." columns in main paper. Trained with only $2.1k$ data, both the 3B and 7B model demonstrate significant performance gains.

**Full Setting Comparison.**    Improved training algorithm leads to improved performance. To better reveal the model performance beyond the limited data scale of Geo3K, we conduct full 30k-sample 150-step experiment on Qwen2.5-VL-7B. As shown in Tab. 10, Shuffle-R1 achieves clear and consistent gains over strong baselines across all six evaluation benchmarks. This result further proven the robustness and scalability of Shuffle-R1.

**Comparison with more RL algorithms.**    We provide additional comparisons with more representative RL algorithms, i.e. RLOO (Ahmadian et al., 2024) and REINFORCE++ (Hu, 2025). We conduct experiments on Qwen2.5-VL-3B-Instruct with Geometry3K dataset. All the experiment settings are kept the same as GRPO/DAPO/GSPO in main paper. The final model performance is

Table 9: Detailed performance on out-of-domain benchmarks of models trained on Geometry3K and K12 data. Highest accuracy marked in **Bold**.

| Method | MathVerse | MathVision | MathVista | WeMath | HallBench | ChartQA | Total Avg. |
|---|---|---|---|---|---|---|---|
| Qwen2.5-VL-3B | 34.77 | 21.94 | 58.40 | 51.72 | 59.83 | 73.08 | 49.96 |
| + Ours (Geo3K) | 43.55 | 25.30 | 61.80 | 64.14 | 63.19 | 77.04 | 55.84 |
| + Ours (K12) | **44.06** | **26.48** | **64.90** | **64.77** | **65.72** | **78.28** | **57.37** |
| Qwen2.5-VL-7B | 42.59 | 25.76 | 67.40 | 63.51 | 65.19 | 79.84 | 57.38 |
| + Ours (Geo3K) | **50.96** | 27.47 | 70.90 | 69.19 | 69.51 | 81.64 | 61.61 |
| + Ours (K12) | 48.59 | **28.61** | **73.20** | **69.71** | **69.87** | **82.60** | **62.09** |

Table 10: Full 30k experiment on Qwen2.5-VL-7B. Highest accuracy marked in **Bold**.

| Method | MathVerse | MathVision | MathVista | WeMath | HallBench | ChartQA | Total Avg. |
|---|---|---|---|---|---|---|---|
| Qwen2.5-VL-7B | 42.6 | 25.8 | 67.4 | 63.5 | 65.2 | 79.8 | 57.4 |
| + GRPO | 50.6 | 28.3 | 74.5 | 69.7 | 70.7 | 81.4 | 62.5 |
| + DAPO | 51.4 | 28.8 | 75.3 | 71.5 | **71.0** | 82.8 | 63.4 |
| + GSPO | 50.8 | 28.2 | 75.3 | 70.1 | 69.7 | 82.9 | 62.8 |
| + Ours | **53.9** | **30.0** | **77.0** | **72.3** | **71.0** | **84.1** | **64.7** |

shown in Tab. 11. Our framework outperform these algorithms by a large margin in both in-domain and out-of-domain tasks.

**Comparison with Prioritize Experience Replay.** We conduct a comparative study by replacing ABS with a prioritized experience replay mechanism. Experience replay maintains a decoupled buffer of past samples, whereas ABS adopts an online, in-place shuffle strategy to dynamically reshape the data distribution. As shown in Fig. 9(a) and (b), in the later stages, the experience replay setting exhibits a plateau in training accuracy and even a drop in validation accuracy, indicating potential overfitting to stale samples. This suggests that prioritized experience replay may overly emphasize historical trajectories, leading to suboptimal convergence. Moreover, ABS proves to be more effective in mitigating the Rollout Silencing.

**Detailed Impact of Hyperparameters.** Here, we provide detailed model performance comparison under different hyperparameter settings in Fig. 12, as supplementary information to corresponding ablation experiments in main paper. For sampling ratio ($\alpha$), both the in-domain and out-of-domain accuracy improves as we increase $\alpha$ from 0.25 to 0.5. The model performance shows a mild decrease when $\alpha$ is further raised to 0.75. This result demonstrates again that not all the rollouts contribute equally positive to RL training. For shuffle times ($S$), the model performance receives a consistent gain when $S$ is increased from 4 to 8, but declines when $S$ is set to 16. This result sug-

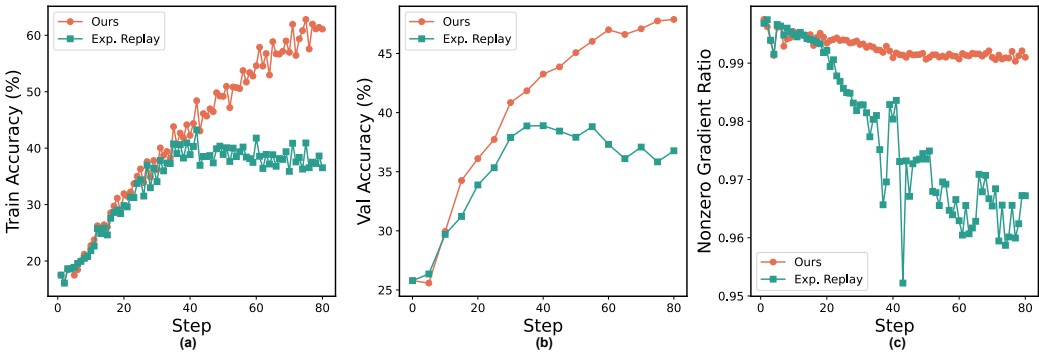

Figure 9: Performance of Shuffle-R1 compared with directly applying Prioritized Experience Replay.

Table 11: Performance of Shuffle-R1 on Geometry3K dataset compared with RLOO and REIN-FORCE++. Highest accuracy marked in **Bold**.

| Method | Geo3K | MathVerse | MathVision | MathVista | WeMath | HallBench | ChartQA |
|---|---|---|---|---|---|---|---|
| Qwen2.5-VL-3B | 25.79 | 34.77 | 21.94 | 58.40 | 51.72 | 59.83 | 73.08 |
| + RLOO | 42.09 | 39.94 | 22.96 | 58.90 | 59.48 | **64.24** | 76.68 |
| + REINFORCE++ | 41.76 | 41.70 | 24.86 | 60.90 | 63.51 | 62.99 | 76.20 |
| + Ours | **47.88** | **43.55** | **25.30** | **61.80** | **64.14** | 63.19 | **77.04** |

Table 12: Performance of Shuffle-R1 under different hyperparameter settings. Highest accuracy marked in **Bold**.

| Setting | Geo3K | MathVerse | MathVision | MathVista | WeMath | HallBench | ChartQA |
|---|---|---|---|---|---|---|---|
| \multicolumn{8}{c}{*Ablation on sampling ratio $\alpha$ ($S = 8$)*} |
| $\alpha = 0.25$ | 45.75 | 42.43 | 23.05 | 60.20 | 63.62 | 63.09 | 75.36 |
| $\alpha = 0.50$ | **47.88** | **43.55** | 25.30 | **61.80** | 64.14 | **63.19** | **77.04** |
| $\alpha = 0.75$ | 47.41 | 41.70 | **25.62** | 60.90 | **64.54** | 62.99 | 73.36 |
| \multicolumn{8}{c}{*Ablation on shuffle times $S$ ($\alpha = 0.5$)*} |
| $S = 4$ | 45.92 | 42.21 | 24.78 | **62.30** | 64.08 | 62.25 | 75.60 |
| $S = 8$ | **47.88** | 43.55 | **25.30** | 61.80 | **64.14** | **63.19** | **77.04** |
| $S = 16$ | 47.23 | **43.70** | 24.86 | 60.90 | 62.64 | 62.20 | 74.84 |

gests that appropriate repeated training on high-quality samples can enhance the model's reasoning ability without affecting data diversity.

## D  EXPERIMENT SETTINGS

We report details of our training and evaluation settings here, including reward function design, main hyperparameters and computing resources.

**Reward Calculation.**  We adopt a combination of format reward and accuracy reward as the final reward in reinforcement learning. The format reward and accuracy reward are calculated as follows:

$$r_{\text{format}} = \begin{cases} 1, & \text{if format is } correct \\ 0, & \text{if format is } incorrect \end{cases} \quad (10)$$

$$r_{\text{acc}} = \begin{cases} 1, & \text{if answer = ground truth} \\ 0, & \text{if answer} \neq \text{ground truth} \end{cases} \quad (11)$$

The final reward is the weighted sum of above rewards:

$$r_{\text{overall}} = 0.1 \times r_{\text{format}} + 0.9 \times r_{\text{acc}} \quad (12)$$

Format reward is assigned to a smaller weight since response formatting is easy to learn.

**Hyperparameters.**  We use EasyR1 (Yaowei et al., 2025) as our training framework. Full hyperparameter settings during training is shown in Tab. 13. For experiments on Geometry3K and K12 with $\sim 2.1k$ training samples, we set the training steps to 80. For the joint training experiments ($\sim 30k$ training samples), we increase the training steps to 150 due to extended data size. Other hyperparameters that are not mentioned are kept to default values of EasyR1.

Table 13: Hyperparameter settings.

| Hyperparameters | Value |
|---|---|
| max pixels | 1000000 |
| min pixels | 262144 |
| max prompt length | 2048 |
| max response length | 2048 |
| rollout batch size | 512 |
| global batch size | 128 |
| learning rate | 1e-6 |
| optimizer | AdamW |
| rollout temperature | 1.0 |
| rollout top p | 0.99 |
| evaluation temperature | 0.5 |
| rollout group number | 16 |
| PTS sampling ratio ($\alpha$) | 0.5 |
| ABS shuffle times ($S$) | 8 |
| KL coefficient | 0.0 |
| vision encoder | frozen |

**Computing Resources.**  All experiments are conducted on $8 \times$ 80G GPUs.

## E    DATASET COLLECTION

For training dataset, we choose Geometry3K (Lu et al., 2021) and MMK12 dataset. Geometry3K dataset is a high quality real-world geometry problem solving dataset, containing 2.1K training samples and 601 test samples. It contains a wide range of geometry problems with varying levels of difficulty. The text problems are very compact with most of the geometric conditions represented in images, making it suitable for our training. MMK12 dataset is introduced by MM-Eureka (Meng et al., 2025), which contains 16k math reasoning training samples. The training samples have both geometric and non-geometric problems and have been carefully examined and manually filtered to ensure quality. The test set of MMK12 further includes other STEM problems such as physics, biology and chemistry. During our training, we used a randomly selected subset of MMK12 provided by NoisyRollout (Liu et al., 2025a) (referred to as 'K12' in the paper to distinguish it from full set of MMK12), which has 2.1k samples with the same size as Geometry3K. All the training samples are in free-form format. Examples of training data shown in Fig. 10.

To fully evaluate model's reasoning ability, apart from in-domain test set from training dataset, we select several representative benchmarks to examine model's performance on math reasoning, visual perception, and chart understanding tasks. For math reasoning task, we choose MathVerse (Zhang et al., 2024b), MathVision (Wang et al., 2024), WeMath (Qiao et al., 2025) and MathVista (Lu et al., 2023). We choose HallusionBench (Guan et al., 2024) and ChartQA (Masry et al., 2022) for visual perception and chart understanding tasks respectively. We believe that a good RL algorithm can not only improve in-domain performance, but also have the potential to promote robust generalization to out-of-domain tasks. Examples of evaluation data shown in Fig. 11.

## F    EVALUATION METRICS

We adopt pass@1 accuracy as evaluation metric. During evaluation, we set the decoding temperature to 0.5 and perform 8 independent runs, reporting the average pass@1 accuracy as final metric. The choice of temperature doesn't degrade model performance, and the averaged accuracy reduces the randomness, resulting in a more stable and reliable evaluation. Prompt format for evaluation is kept identical to training prompt. Evaluation settings of our model and all the reproduced results in main paper are kept the same to ensure a fair comparison.

For evaluation of MathVerse, MathVision, WeMath, MathVista and ChartQA, we employ Gemini-2.0-Flash-001 (Deepmind, 2025) to first extract the predicted answer from model response then compare it with ground truth. Fig. 12 and Fig. 13 demonstrate the extraction and verification prompt

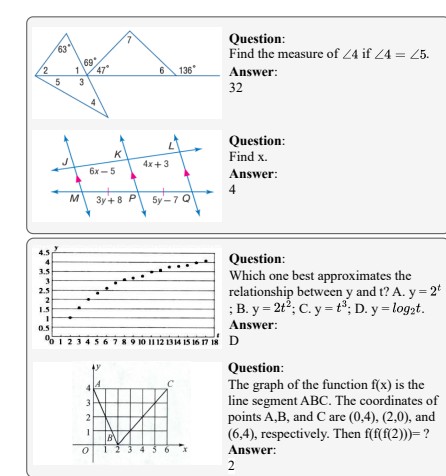

Figure 10: **Top**: Examples of Geometry3K. **Bottom**: Examples of MMK12.

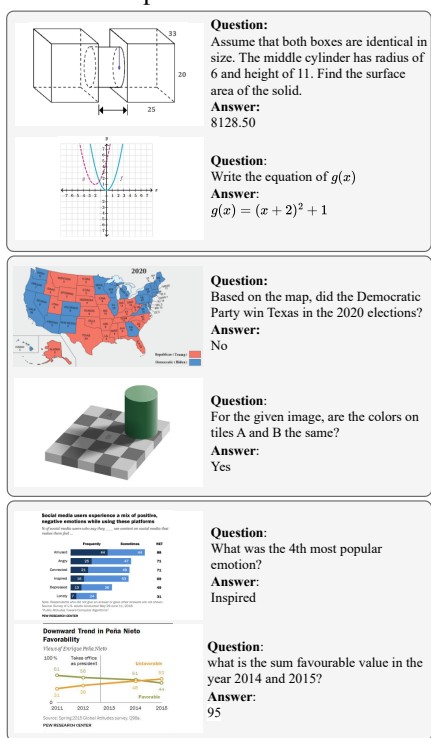

Figure 11: **Top**: Examples of math reasoning tasks. **Middle**: Examples of visual perception tasks. **Bottom**: Examples of chart understanding tasks.

Figure 12: Extraction prompt for Gemini.

Figure 13: Score prompt for Gemini.

for Gemini. Specifically, we report accuracy on WeMath under loose mode, and overall accuracy on MathVerse (including all sub categories: Text Dominant, Text Lite, Vision Intensive, Vision Dominant and Vision Only).

## G CASE STUDY

Here we provide some qualitative case study of Shuffle-R1's reasoning outputs.

Fig. 14 and Fig. 15 show two improved cases on math related tasks. In case 1, the base model has an error in visual information parsing, referring to angles not existed in the figure, resulting in a reasoning fault. The RL model correctly parsed and solved the geometry problem. In case 2, the base model misused a geometric theorem, leading to wrong answer, while the RL model correctly solved the problem with accurate theorem.

Fig. 16 demonstrates improved reasoning ability in RL model, where the base model has accurate perception about the chart but has reasoning error in CoT. The RL model can not only accurately parse the visual information, but also perform correct reasoning to finally reach the right answer.

Case 4 in Fig. 17 further demonstrates that RL model also has better visual perception ability compared to base model. The figure from HallusionBench has been artificially inserted with an image of a hen, which only occupies a very small region of the original image. This modification has resulted in a perception error and in base model, but the RL model can accurately identify the inserted image.

## H THEORETICAL ANALYSIS OF SHUFFLE-R1

In this section, we provide an intuitive analysis on how the adaptive selection & resampling improve the training dynamics, and discuss the bias introduced by this process. We begin by assuming the Advantage($A$) follows a Gaussian distribution, $A \sim \mathcal{N}(0, \sigma^2)$, which matches the observation of our probe analysis.

We define the original expectation of $|A|$ as:

$$E_{\text{orig}} = \mathbb{E}[|A|]. \tag{13}$$

We simplify the PTS as a truncate process with a threshold $\tau > 0$, and define the kept set $S^{\text{PTS}} = \{|A| > \tau\}$, the corresponding conditional expectation is:

$$E_{\text{PTS}} = \mathbb{E}[|A| \mid |A| \geq \tau]. \tag{14}$$

On the truncated set, we sample with weights $w(a) \propto |a|$, and the resulting expectation is denoted as:

$$E_{\text{new}} = \mathbb{E}[w(a)|a| \mid |a| \in S^{\text{PTS}}]. \tag{15}$$

## H.1 Intuitive Analysis on Training Dynamics

**Proposition 1 (Amplification on expectation of Advantage).**

*Proof.* Define $X = |A|$ on the $S^{\text{PTS}}$, the weighted sampling probability density can be expressed as:

$$p_{\text{new}}(x) = \frac{x \cdot p(x \mid S^{\text{PTS}})}{\mathbb{E}[X \mid S^{\text{PTS}}]} = \frac{x \cdot p(x \mid S^{\text{PTS}})}{E_{\text{PTS}}}. \tag{16}$$

Thus the weighted sampling expectation can be calculated as:

$$E_{\text{new}} = \int x \cdot \frac{x \cdot p(x \mid S^{\text{PTS}})}{E_{\text{PTS}}} dx = \frac{\mathbb{E}[X^2 \mid S^{\text{PTS}}]}{\mathbb{E}[X \mid S^{\text{PTS}}]} = \frac{E_{\text{PTS}}^{(2)}}{E_{\text{PTS}}}. \tag{17}$$

Based on Cauchy–Schwarz inequality, we can derive that:

$$\mathbb{E}[X^2] \geq (\mathbb{E}[X])^2 \implies E_{\text{new}} \geq E_{\text{PTS}} \tag{18}$$

Further, the original expectation can be decomposed as:

$$E_{\text{orig}} = (1 - p_\tau)\mathbb{E}[|A| \mid |A| < \tau] + p_\tau \mathbb{E}[|A| \mid |A| \geq \tau], \quad \text{where } p_\tau = \mathbb{P}(|A| \geq \tau). \tag{19}$$

Since

$$\mathbb{E}[|A| \mid |A| < \tau] < \tau < \mathbb{E}[|A| \mid |A| \geq \tau]. \tag{20}$$

we have $E_{\text{PTS}} > E_{\text{orig}} \implies E_{\text{new}} > E_{\text{orig}}$. $\square$

**Proposition 2 (Amplification on gradient norm).**

*Proof.* In the policy gradient training, the per-sample gradient can be expressed as:

$$g = A \cdot u. \qquad u = \nabla_\theta \log \pi_\theta(o). \tag{21}$$

$$||g|| = |A| \, ||u||. \tag{22}$$

Derived from the standard assumptions of policy gradient that $||u||$ is approximately independent from $|A|$, we have:

$$\mathbb{E}[||g_{\text{new}}||] \propto \mathbb{E}_{\text{new}}[|A|] \cdot \mathbb{E}[||u||] > \mathbb{E}_{\text{orig}}[|A|] \cdot \mathbb{E}[||u||] = \mathbb{E}[||g_{\text{orig}}||]. \tag{23}$$

which in turn yields larger update magnitudes per optimization step. $\square$

## H.2 Bias Under Selective Sampling

**Proposition 3 (Adaptive sampling introduce positive bias).**

*Proof.* Under the Gaussian distribution assumption, the expectation can be calculated as:

$$\begin{aligned}
\mathbb{E}_{\text{new}}[A] &= \frac{1}{Z}\left(\int_{-\infty}^{-\tau} a|a|p(a)da + \int_{\tau}^{\infty} a|a|p(a)da\right) \\
&= \frac{2}{Z}\int_{\tau}^{\infty} a^2 p(a)da > 0
\end{aligned}$$

where the normalized factor $Z$ is defined as:

$$\begin{aligned}
Z &= \int_{-\infty}^{\infty} |a|p(a)\mathbb{I}(|a| > \tau)da \\
&= 2\int_{\tau}^{\infty} a \cdot p(a)da
\end{aligned}$$

The positive bias indicates that the gradient update is systematically tend to trajectories that has higher advantage. $\square$

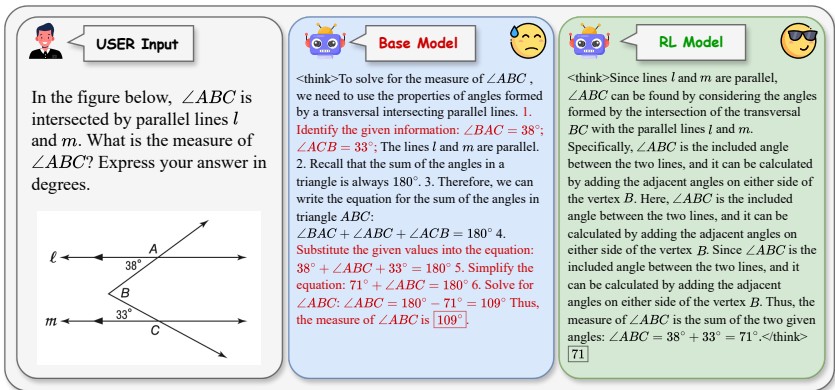

Figure 14: A case study on math related task.

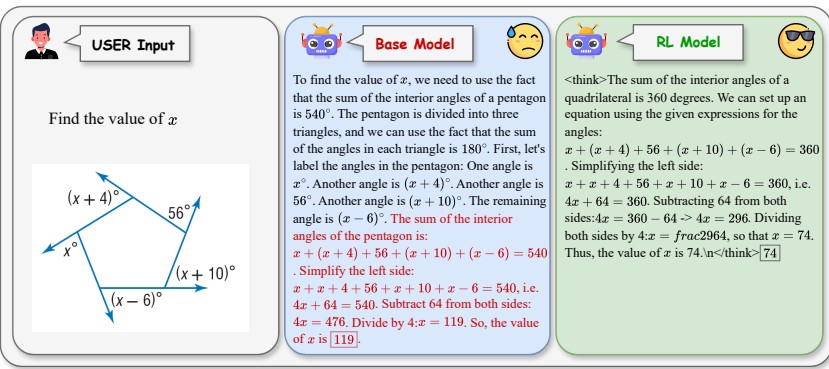

Figure 15: Another case study on math related task.

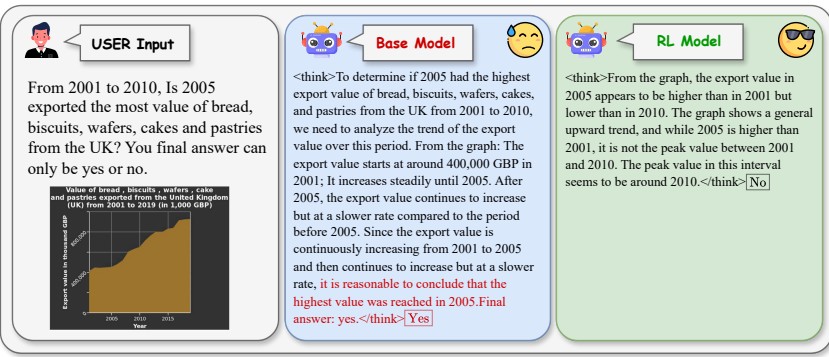

Figure 16: A case study on chart understanding task.

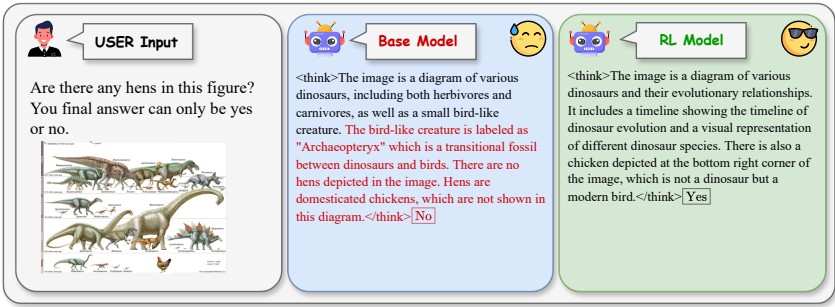

Figure 17: A case study on visual perception task.

