# OpenReview forum: "Shuffle-R1: Efficient RL framework for Multimodal Large Language Models via Data-centric Dynamic Shuffle"
_ICLR.cc/2026/Conference — ICLR 2026 Poster_

### Official Review · Reviewer_u5Bq · 2025-10-20

**Soundness:** 3
**Presentation:** 3
**Contribution:** 2
**Rating:** 4
**Confidence:** 4

**Summary:**

This paper identifies two key inefficiencies in reinforcement learning (RL) fine-tuning for multimodal large language models (MLLMs): Advantage Collapsing—where most advantage estimates cluster near zero, weakening gradient signals—and Rollout Silencing—where fewer rollouts contribute useful gradients as training progresses. To address these, the authors propose Shuffle-R1, a data-centric RL framework featuring: (1) Pairwise Trajectory Sampling, which forms high-contrast trajectory pairs by matching high- and low-advantage rollouts to amplify informative signals, and (2) Advantage-based Batch Shuffle, which dynamically reshuffles training batches to prioritize high-value samples based on their advantage magnitudes. Experiments show Shuffle-R1 consistently outperforms strong baselines (e.g., GRPO, DAPO) across multiple multimodal reasoning benchmarks, achieves competitive results with leading closed-source models like GPT-4o and Claude-3.7, and even matches prior methods using only half the training steps—all with minimal computational overhead. The framework also generalizes to text-only LLMs, highlighting its broad applicability.

**Strengths:**

It demonstrates high quality through rigorous experiments across model scales, datasets, and multimodal benchmarks, supported by thorough ablations and efficiency analyses.
The writing is clear: concepts are intuitively explained, methods are well-structured, and figures effectively illustrate key ideas.

**Weaknesses:**

Unfair rollout budget: The method uses 16 rollouts per query but only trains on 8, while baselines like DAPO likely use only 8 total. The gains may come from more exploration, not smarter sampling. A fair comparison—using the same total number of rollouts (e.g., 16 for both)—is missing. Also, as N grows large, discarding (1−α) samples wastes potentially useful signals.

Resampling vs. reweighting: The paper uses advantage-based resampling (ABS), but reweighting the loss by advantage magnitude would achieve nearly the same effect with zero extra overhead and less variance. No justification or comparison is provided for choosing the more complex resampling approach.

Marginal gains vs. added cost: Improvements over strong baselines (e.g., DAPO, GSPO) are small—often <1% on average—yet the method adds rollout generation cost (2× more rollouts) and pipeline complexity. If the real benefit is faster convergence (e.g., 2× fewer steps), that should be the highlighted advantage, not final accuracy. As-is, the practical value is unclear.

**Questions:**

see weakness

---

> ### Author Response · Authors · 2025-11-20
> **Response to Reviewer u5Bq (1 of 2)**
>
> We appreciate your valuable feedback. Below are our responses to your concerns.
>
> ## R.W1
>
> **About Unfair Comparison**
>
> Exploration indeed plays a critical role in RL, but we'd like to clarify that simply increasing the rollout size does not address the Advantage Collapsing and Rollout Silencing issue. Shuffle-R1 dynamically selects and resamples the training batch, reshapes the data distribution for better RL training.
>
> We conducted an additional experiment to strictly align the rollout budget. The baseline methods use 2N=16 rollout size and randomly prune $\alpha=0.5$ of the total rollouts. The results are shown in the table below:
>
> | Qwen-2.5-VL-3B | Geo3K | Math Avg. | HallusionBench| ChartQA |
> | --- | --- | --- | --- | --- |
> | GRPO-align | 42.09 | 45.66 | 62.98 | 76.36 |
> | DAPO-align | 45.28 | 48.10 | 63.32 | 76.64 |
> | GSPO-align | 43.43 | 47.81 | 63.82 | 75.52 |
> | Ours | 47.88 | 48.70 | 63.19 | 77.04 |
>
> All the baselines show no performance gain even doubling the rollout budget. The results show that performance gain of Shuffle-R1 is not simply benefit from a bigger rollout budget, but from a better data distribution refined by combined effect of PTS and ABS.
>
> **About Over-pruning**
>
> We agree that discarding rollouts could theoretically reduce the amount available signals. However, as discussed in the paper, our probe analysis shows that most rollouts lie close to "zero advantage area". By strategically setting $\alpha$, we can reduce the negative impact of over-pruning.
>
> Furthermore, the proportional pruning used by Shuffle-R1 is scale-invariant, under the same probability distribution, increasing the rollout size will not cause useful high-signal samples to be mis-pruned.
>
> To verify this, we conduct an additional experiment to verify the impact of rollout size. We scale the rollout size to 2N=24, with update size=12. The results are shown in the table below.
>
> | Qwen-2.5-VL-3B | Geo3K | Math Avg. | HallusionBench| ChartQA |
> | --- | --- | --- | --- | --- |
> | GRPO | 43.76 | 46.9 | 63.98 | 76.12 |
> | Shuffle-R1 | 49.75 | 49.66 | 63.30 | 77.98 |
>
> Shuffle-R1 still delivers superior performance, suggesting that the effect of over-pruning is not as significant as expected.
>
>
> ## R.W2
> Thank you for raising this concern. Although advantage-based reweighting may appear mathematically similar to resampling, it lacks the ability to  proactively adjust and reshape the data distribution during training. To verify their effectiveness, we conducted an additional experiment to compare the performance of Reweighting and Shuffle-R1. We replace ABS with a softmax reweighting process based on the absolute advantage. The results are shown in the table below.
>
> | Qwen-2.5-VL-3B | Geo3K | Math Avg. | HallusionBench| ChartQA |
> | --- | --- | --- | --- | --- |
> | Reweight | 46.75 | 47.98 | 63.51 | 75.39 |
> | Shuffle-R1 | 47.88 | 48.70 | 63.19 | 77.04 |
>
> Reweighting consistently underperforms Shuffle-R1. This confirms that a better sample distribution cannot be obtained simply by resampling. In contrast, resampling directly reshapes the data distribution in a bidirectional manner: (1) it directly reduces the number of zero-gradient samples, avoiding useless calculation and updates; (2) it fills the vacancy with high-advantage samples, allowing the model to fully grasp the optimal response pattern.
>
> By integrating ABS in the training process, Shuffle-R1 directly refines the data distribution into a well-spread one, overcoming the Rollout Silencing issue. This effect cannot be achieved simply by reweighting.

---

> ### Author Response · Authors · 2025-11-20
> **Response to Reviewer u5Bq (2 of 2)**
>
> ## R.W3
> Thank you for your thoughtful comment. We would like to clarify that the practical value of Shuffle-R1 lies in two aspects: (1) faster training convergence, and (2) stronger model performance.
>
> Shuffle-R1 achieves faster convergence due to better organized data distribution by dynamic and adaptive data structuring. It reaches performance comparable to full-step GRPO with approximately half of the training step and 60% wall-clock time (Fig. 7 in Section 4.2). For direct comparison, we additionally evaluate 40-step early-stop performance.
>
> | Qwen-2.5-VL-3B (2.1k data) | Geo3K | Math Avg. | HallusionBench| ChartQA | Relative Time |
> | --- | --- | --- | --- | --- | --- |
> | GRPO (40-step) | 38.76 | 44.19 | 60.61 | 75.36 | 0.50x |
> | DAPO (40-step) | 39.76 | 45.54 | 61.24 | 75.72 | 0.59x |
> | **Shuffle-R1 (40-step)** | **42.24** | **46.82** | **63.41** | **76.32** | **0.53x** |
> | **GRPO (80-step)** | **42.64** | **46.74** | **63.09** | **76.20** | **1.00x** |
> | DAPO (80-step) | 45.09 | 48.08 | 63.24 | 76.70 | 1.17x |
> | Shuffle-R1 (80-step) | 47.88 | 48.70 | 63.19 | 77.04 | 1.04x |
>
> Shuffle-R1 consistently outperforms baseline by a large margin, demonstrating more efficient reward signals utilization and reduce in training budget.
>
> Beyond faster convergence, the improved training algorithm leads to improved performance. To better reveal the model performance beyond the limited data scale of Geo3K, we conduct full 30k-sample 150-step experiment on Qwen2.5-VL-7B.
>
> | Qwen-2.5-VL-7B (30k data) | MathVerse | MathVision | MathVista | WeMath | HallusionBench| ChartQA | Avg. |
> | --- | --- | --- | --- | --- | --- | --- | --- |
> | GRPO | 50.6 | 28.3 | 74.5 | 69.7 | 70.7 | 81.4 | 62.5 |
> | DAPO | 51.4 | 28.8 | 75.3 | 71.5 | 71.0 | 82.8 | 63.4 |
> | GSPO | 50.8 | 28.2 | 75.3 | 70.1 | 69.7 | 82.9 | 62.8 |
> | Shuffle-R1 | 53.9 | 30.0 | 77.0 | 72.3 | 71.0 | 84.1 | 64.7 |
>
> Shuffle-R1 achieves clear and consistent gains over strong baselines across all evaluation benchmarks, confirming that its benefit is neither marginal nor dataset-specific, but robust and scalable.
>
> In summary, Shuffle-R1 achieves faster convergence and improves model performance. It utilizes data and compute more effectively without introducing substantial overhead. We believe the idea of online dynamic data structuring offers potential value for the development of RL post-training.

---

> ### Author Response · Authors · 2025-11-26
> **Looking forward to hearing from you**
>
> Dear reviewer `u5Bq`,
>
> We are grateful for your constructive comments on our submission.
>
> In response to your review, we have added extensive experiments to clarify and answer your questions and concerns. We made fully-aligned experiment to better clarify our improvement and higher $N$ experiment to prove the minimal effect of potential over-pruning. We also made reweighting experiments to show our method's effectiveness over the reweighting strategy. In addition, we made a clearer and comprehensive discussion on the practical value of Shuffle-R1 in improving convergence speed and model performance at the same time.
>
> Your feedback and suggestions are very instructive in refining and strengthening our work. We sincerely appreciate your effort again.
>
> As we are now in the Author–Reviewer Discussion phase, we would be grateful for any additional comments you may wish to provide as the discussion period progresses.
>
> Looking forward to hearing from you again.
>
> Best regards,
>
> The Authors of Submission 8742

---

> > ### Comment · Reviewer_u5Bq · 2025-11-27
> > **Official Comment**
> >
> > Thank authors for their detailed response. I have adjusted my score accordingly.

---

> > > ### Author Response · Authors · 2025-11-27
> > > **We appreciate your acknowledgement**
> > >
> > > Dear reviewer `u5Bq`,
> > >
> > > We sincerely appreciate your recognition of our work. Thank you again for your efforts!
> > >
> > > Best regards,
> > >
> > > The Authors of Submission 8742

---

### Official Review · Reviewer_gi6w · 2025-10-28

**Soundness:** 3
**Presentation:** 3
**Contribution:** 4
**Rating:** 6
**Confidence:** 3

**Summary:**

This paper proposes Shuffle-R1, a data-centric reinforcement learning framework that targets two major issues in current RL fine-tuning pipelines: Advantage Collapsing (most advantages cluster near zero) and Rollout Silencing (the share of rollouts with non-zero gradients keep dropping).

The method introduces two simple modules: Pairwise Trajectory Sampling, which selects high-contrast trajectory pairs to strengthen gradient signals, and Advantage-based Batch Shuffle, which dynamically reshuffles batches to reuse more informative samples.

Experiments across multimodal reasoning benchmarks show consistent gains over GRPO, DAPO, and GSPO with almost no extra computational cost.

**Strengths:**

1. The paper pinpoints two concrete and observable issues, "Advantage Collapsing and Rollout Silencing", which intuitively explain why current RL pipelines waste computation and fail to leverage informative signals. This diagnostic perspective is well-motivated.

2. Instead of modifying the reward model or policy objective, Shuffle-R1 improves RL efficiency purely from the data side through Pairwise Trajectory Sampling (PTS) and Advantage-based Batch Shuffle (ABS). Both modules are lightweight, easy to implement, and can plug into existing frameworks without architectural changes. And the proposed framework shows consistent improvements across different datasets, model scales, and reasoning benchmarks with less training steps and GPUs.

3. The paper presents clear ablation studies and analyses showing how PTS mitigates advantage collapse and how ABS maintains token utilization over time. This makes the method’s effectiveness both transparent and reproducible.

**Weaknesses:**

1. **Overfitting to high-advantage samples**
Since ABS repeatedly exposes high-value trajectories, the framework might bias the model toward a narrower distribution of “reward-dense” samples, reducing exploration and long-term diversity.

2. **Scope of benchmarks**
Most experiments are on math or visual reasoning tasks; while results are strong, these domains already have dense reward signals. It remains unclear whether Shuffle-R1 would bring similar benefits on tasks with sparse or noisy rewards (e.g. open-ended QA, safety...)

3. **Lack of comparison to recent adaptive-sampling paradigms**
Recent works like LIMO also tackle signal efficiency through adaptive data selection and contrastive training. Including these would further enhance impact within the community.

**Questions:**

Overall, this paper provides a well-motivated and practically useful data-centric perspective on improving RL efficiency for multimodal LLMs, there are several questions:
1. Since ABS repeatedly exposes high-advantage trajectories, how does the framework prevent over-exploitation of a narrow subset of rollouts?

2. The experiments are convincing but domain specific. Does the author expect the same efficiency gains for RL tasks with sparse, delayed, or non-verifiable rewards (e.g., open-ended QA)? If not, what modifications would be necessary?

---

> ### Author Response · Authors · 2025-11-20
> **Response to Reviewer gi6w**
>
> Thank you for your valuable feedback. Here are our responses to your questions and concerns.
>
> ## R.W1 & Q1
> Thank you for raising this concern. During the design of Shuffle-R1, we paid special attention to the potential risk of overfitting. To be specific, the algorithm is designed to mitigate overfitting through the following mechanisms:
>
> 1. Controllable Replay Intensity: Shuffle-R1 does not repeatedly focus on only a few rollouts. For each sub-batch sample, we use non-replacement sampling, ensuring each element in th sub-batch is unique. Shuffle Times $S$ controls the total number of sub-batch sample. Under this setting, each trajectory can only be replayed $S$ times at most.
> 2. Low-Advantage Aware Soft Sampling: Instead of hard filtering, ABS uses weighted probabilistic soft sampling, allowing low-advantage samples to still be sampled at a lower probability. This prevents mode collapse and maintains exploration, rather than collapsing the distribution toward exclusively high-advantage samples.
> 3. Dynamic Sampling Strategy: Advantages shift as the policy improves. As a result, ABS reallocates focus toward currently informative trajectories rather than repeatedly exploiting a fixed subset. This dynamic reallocation naturally guards against overfitting to a static slice of data.
>
> In the Appendix C of the paper, we have compared Shuffle-R1 with vanilla Prioritize Experience Replay. The Prioritize Experience Replay setting plateaus early during training, indicating a potential overfitting. In contrast, Shuffle-R1 maintains a stable performance throughout the training process.
>
>
> ## R.W2 & Q2
> We appreciate your suggestion. Shuffle-R1 primarily focuses on outcome reward RL in sparse reward scenarios, but its algorithmic design and code implementation are fully compatible with dense reward settings. To investigate the generalization ability of our algorithm on RL scenarios beyond sparse reward signals, we conducted experiments on the Referring Expression Comprehension (REC) task, a subset of Visual Grounding that adopts IoU-based soft rewards.
>
> We follow VLM-R1[1] and trained Qwen2.5-VL-3B using 60k samples randomly sampled from the RefCOCO/RefCOCOg/RefCOCO+ datasets for 500 steps. The detailed experimental results are presented below
>
> | Model | RefCOCO testA | RefCOCO testB | RefCOCO+ testA | RefCOCO+ testB | RefCOCOg test |
> | --- | --- | --- | --- | --- | --- |
> | Base* | 86.09 | 75.64 | 81.71 | 66.93 | 72.39 |
> | GRPO | 89.90 | 81.33 | 85.94 | 70.97 | 81.45 |
> | Shuffle-R1 | 91.83 | 84.31 | 87.84 | 76.27 | 86.07 |
>
> *: Base model uncontrollably outputs boxes in different formats (absolute, 0-1 relative, or 0-1000 normalized). We transform the detected 0-1 relative boxes to absolute boxes when computing IoU, resulting in lower accuracy on some test set compared to official tech report.
>
> The results show that Shuffle-R1 also exhibits strong generalization on the REC task. Since other tasks can either be transformed into a sparse reward scenario (like the base setting in our paper) or a soft reward scenario (like the REC task), we believe the results provide a representative evaluation of the generalizability of our algorithm.
>
>
> ## R.W3
> Thank you for the suggestion. Here we compare and discuss our method with LIMO[2] and following research LIMR[3].
>
> LIMO assumes the base model already encodes domain knowledge and uses a multi-stage data curation pipeline to construct 800 high-quality CoT traces. By performing SFT on this dataset, LIMO achieves significant performance gain on math reasoning tasks. Shuffle-R1, on the other hand, focuses more on dynamic optimization of MLLM RL post training. The two methods are orthogonal and have the potential to complement each other. However, later research points out that LIMO only introduces limited gains for 7B-scale models.
>
> LIMR introduces the core idea of LIMO into RL training. By leveraging a specifically designed difficulty and quality evaluation/filter process, LIMR achieves better performance using only 1000 samples than other methods with more than 8k samples.
>
> Using the same base model (Qwen2.5-Math-7B), we compare LIMR and Shuffle-R1 on various reasoning benchmarks. The results are shown in the table below.
>
> | Qwen2.5-Math-7B | Math12K | AIME24 | MATH500 | GSM8K | GPQA Diamond | OlympiadBench |
> | --- | --- | --- | --- | --- | --- | --- |
> | LIMR | 76.6 | 23.3 | 78.2 | 91.4 | 36.3 | 38.9 |
> | Shuffle-R1 | 78.2 | 23.3 | 79.4 | 89.5 | 37.3 | 41.4 |
>
> Shuffle-R1 outperforms LIMR on most benchmarks, demonstrating the effectiveness of our method.
>
>
> ### References：
> [1] VLM-R1: A Stable and Generalizable R1-style Large Vision-Language Model, arXiv:2504.07615
>
> [2] LIMO: Less Is More For Reasoning, arXiv:2502.03387
>
> [3] LIMR: Less Is More For RL Scaling, arXiv:2502.11886

---

> ### Author Response · Authors · 2025-11-26
> **Looking forward to hearing from you**
>
> Dear reviewer `gi6w`,
>
> Thank you again for your thoughtful and constructive review.
>
> In response to your questions, we have made in-depth clarifications on how we prevent the potential overfitting issue in Shuffle-R1. To further investigate the generalization capability of Shuffle-R1 beyond visual reasoning tasks, we added experiment on Referring Expression Comprehension task. What's more, we made further discussion about LIMO and LIMR and provided a performance comparison on Qwen2.5-Math-7B with LIMR as requested.
>
> Your feedback and suggestions are very instructive in refining and strengthening our work. We sincerely appreciate your effort again.
>
> As we are now in the Author–Reviewer Discussion phase, we are welcome to any further discussion you may have during the process.
>
> Looking forward to hearing from you again.
>
> Best regards,
>
> The Authors of Submission 8742

---

> > ### Comment · Reviewer_gi6w · 2025-11-26
> >
> > Thank you for the additional experiments and clarifications. The new results on REC task, and the comparison with LIMO/LIMR address most of my concerns. These updates clearly strengthen the paper and further validate the effectiveness of your method.
> >
> > Overall, this is a solid and impressive work. I’m happy to raise my score.

---

> > > ### Author Response · Authors · 2025-11-26
> > > **Thank you for your acknowledgment**
> > >
> > > Dear reviewer `gi6w`,
> > >
> > > We sincerely appreciate your feedback, and we are glad to hear that our additional experiments and clarifications addressed your concerns. Thank you again for your positive recognition of our work!
> > >
> > > Best regards,
> > >
> > > The Authors of Submission 8742

---

### Official Review · Reviewer_Codo · 2025-11-01

**Soundness:** 3
**Presentation:** 3
**Contribution:** 3
**Rating:** 6
**Confidence:** 4

**Summary:**

Shuffle-R1 is a data-centric RL fine-tuning method for multimodal LLMs that combats Advantage Collapsing and Rollout Silencing. It introduces Pairwise Trajectory Sampling to extract high-contrast advantage pairs and Advantage-based Batch Shuffle to over-sample valuable pairs during mini-batch construction. On 2–30 k training samples, 3/7 B models outperform GRPO, DAPO, GSPO and match GPT-4o/Claude-3.7 on MathVerse, MathVista, etc.

**Strengths:**

- Diagnose Advantage Collapsing & Rollout Silencing in MLLM-RL; proposes contrastive pairing + advantage-weighted reshuffle instead of larger rollouts or reward re-design.
- Extensive ablations (α, S, PTS variants), 8 datasets, 2 model scales; statistical gains significant; extend to LLMs; code & pseudo-code provided.

**Weaknesses:**

- While the empirical results are strong, the paper lacks formal analysis or theoretical justification for why PTS and ABS improve training dynamics. For example, it would be helpful to show (even intuitively) how contrastive sampling improves gradient variance or convergence rates.
- While Shuffle-R1 outperforms GRPO, DAPO, and GSPO, it does not compare with other data-centric RL methods such as curriculum-based sampling, which are relevant to the idea of reusing or reweighting data. A short discussion or comparison would strengthen the positioning of the work.
- While ablations on α and S are provided, other design choices (e.g., max-min pairing, absolute advantage weighting) are not thoroughly explored. For example, would cosine similarity or entropy-based weighting perform better?

**Questions:**

- Have you empolyed Shuffle-R1 on larger models (e.g., 30B+ parameters) or other domains beyond math and vision reasoning? What are the anticipated challenges?
- Can you provide examples where Shuffle-R1 underperforms or fails to improve over baselines? What are the limitations of your method in its current form?
- Can you provide a more formal or intuitive explanation of how PTS improves gradient estimation? For example, how does selecting high-contrast pairs reduce gradient variance or improve signal-to-noise ratio?

---

> ### Author Response · Authors · 2025-11-20
> **Response to Reviewer Codo (1 of 2)**
>
> We appreciate your valuable review. Here is our response to your concerns and questions.
>
> ## R.W1 & Q3
>
> Thanks for your valuable advice. It is important to theoretically justify the effect of Shuffle-R1 for clearer understanding. We provide the analysis of the change in gradient norm to approximately explain the improvement in training dynamics. Full analysis in Appendix H.1 of the revised paper.
>
> Assuming the original distribution of Advantage follows gaussian distribution $A \sim \mathcal{N}(0, \sigma^2)$ (consistent with probe analysis in Fig. 1), we approximate the PTS process using truncation with threshold $\tau>0$, the retained set can be expressed as $S^{\text{PTS}}=\{ |A|>\tau \}$. Define the following variables:
>
> $E_{\text{orig}} = \mathbb{E}[|A|] = (1 - p_{\tau}) \mathbb{E}[|A|||A|<\tau] + p_{\tau}\mathbb{E}[|A|||A| \geq \tau], \quad \ p_{\tau}=\mathbb{P}(|A| \geq \tau).$
>
> $E_{\text{PTS}}= \mathbb{E}[|A|| |A| \geq \tau]$
>
> $E_{\text{new}}= \mathbb{E}[w(a)|a|||a| \in S^{\text{PTS}}]$
>
> We can deduce:
>
> $E_{\text{new}} = \int x \cdot \frac{x \cdot p(x|S^{\text{PTS}})}{E_{\text{PTS}}} dx = \frac{\mathbb{E}[X^2|S^{\text{PTS}}]}{\mathbb{E}[X|S^{\text{PTS}}]} = \frac{E_{\text{PTS}}^{(2)}}{E_{\text{PTS}}}$.
>
> By Cauchy–Schwarz inequality, $E_{\text{new}} \geq E_{\text{PTS}}$. Further, $E_{\text{PTS}}>E_{\text{orig}}$ because of the truncation. Therefore, $E_{\text{new}} > E_{\text{orig}}$.
>
> In policy gradient training, the per-sample gradient $g=A \cdot u,\ u=\nabla_\theta \text{log}\pi_\theta (o)$, the gradient norm can be expressed as $||g||=|A| \cdot ||u||$. Since $||u||$ is approximately independent from $|A|$, we have:
>
> $\mathbb{E}[||g_{\text{orig}}||]=\mathbb{E}_{\text{orig}}[|A|] \cdot \mathbb{E}[||u||]$
>
> $\mathbb{E}[||g_{\text{new}}||] = \mathbb{E}_{\text{new}}[|A|] \cdot \mathbb{E}[||u||]$
>
> $\mathbb{E}[||g_{\text{new}}||] > \mathbb{E}[||g_{\text{orig}}||]$.
>
>
> This indicates Shuffle-R1 yields larger update magnitudes and stronger training signal per optimization step. It effectively relieves the Rollout Silencing issue where gradient signals fade during training, and improves convergence dynamics under the same compute budget.
>
>
> ## R.W2
> Thank you for pointing this out. Curriculum-based sampling is indeed a major part of data-centric RL, and we agree that situating Shuffle-R1 relative to these methods is necessary.
>
> Curriculum RL methods typically rely on semi-static difficulty estimation and require substantial human-designed preprocessing. Representative examples include:
>
> - ADCL[1], which split the training data into staged batches and periodically recalculates difficulty scores for samples in each stage.
> - E2HReasoner[2], which partitions data by difficulty and uses cosine/Gaussian probability schedules to balance exploration across stages.
> - DUMP[3], which samples from difficulty buckets based on UCB estimates computed online.
>
> In contrast, Shuffle-R1 is fully dynamic and difficulty-free. It does not require any difficulty definition, preprocessing, or curriculum schedule. By adaptively evaluate the significance of samples in the training batch, Shuffle-R1 adaptively selects which samples to update on. The training schedule is much simpler but very effective.
>
> We will add a short discussion in the revised version to better compare Shuffle-R1 with curriculum-based methods.
>
> ### References：
> [1] Learning Like Humans: Advancing LLM Reasoning Capabilities via Adaptive Difficulty Curriculum Learning and Expert-Guided Self-Reformulation, EMNLP 2025.
>
> [2] CURRICULUM REINFORCEMENT LEARNING FROM EASY TO HARD TASKS IMPROVES LLM REASONING, arXiv:2506.06632.
>
> [3] DUMP: Automated Distribution-Level Curriculum Learning for RL-based LLM Post-training, arXiv:2504.09710v1.

---

> ### Author Response · Authors · 2025-11-20
> **Response to Reviewer Codo (2 of 2)**
>
> ## R.W3
> Thank you for your suggestion. In fact, we have conducted extensive explorations on the design of components in Shuffle-R1, aiming to develop a concise yet effective implementation. We supplement these experimental results in the table below.
>
> For Sampling Weight computation, we tried: 1. Cosine Similarity Sampling, calculating the cosine similarity of each rollout in the group and select the most diverse ones. 2. Entropy Sampling, calculating the entropy of each rollout in the group and select ones with the lowest entropy.
>
> For Absolute Advantage Sampling, alongside the current implementation, we also tried: 1. Softmax Weight, calculating the sampling weight using softmax function. 2. Log-Softmax Weight, calculating the sampling weight using log-softmax function.
>
> | Qwen-2.5-VL-3B | Geo3K | Math Avg. | HallusionBench| ChartQA |
> | --- | --- | --- | --- | --- |
> | GRPO | 42.64 | 46.74 | 63.09 | 76.20 |
> | Cosine | 45.92 | 48.08 | 63.51 | 74.72 |
> | Entropy | 45.75 | 47.98 | 63.43 | 74.72 |
> | Softmax | 47.32 | 48.59 | 63.62 | 76.94 |
> | Log-Softmax| 47.25 | 48.10 | 63.32 | 74.64 |
> | Ours | 47.88 | 48.70 | 63.19 | 77.04 |
>
> As shown in the table, though effective, the Cosine Similarity Sampling and the Entropy Sampling do not outperform the current Absolute Advantage Sampling. Softmax Weight and Log-Softmax Weight also yield similar results. Considering both effectiveness and simplicity, we ultimately opted for the straightforward yet most effective Absolute Advantage Sampling with linear weighting.
>
> ## R.Q1
> The scalability and generalizability is significant research directions for a RL algorithm. Our code is directly compatible with the training of 32B models. We have made an additional experiment to verify the effectiveness of Shuffle-R1 on 32B model. We train Qwen2.5-VL-32B on proposed 30K data for 50 steps. The results are shown in the table below.
>
> |Qwen2.5-VL-32B | MathVerse | MathVision | MathVista | WeMath | HallusionBench | ChartQA | Avg. |
> | --- | --- | --- | --- | --- | --- | --- | --- |
> | Base | 57.0 | 38.2 | 75.4 | 72.9 | 71.3 | 80.7 | 65.9 |
> | GRPO | 58.4 | 39.3 | 77.3 | 75.0 | 70.3 | 83.8 | 67.4 |
> | Shuffle-R1 | 59.0 | 41.2 | 79.5 | 77.9 | 72.2 | 84.9 | 69.1 |
>
> Note: We report 50-step results solely for efficiency and comparison purposes. This does not reflect final algorithm performance, which would require much more extended training.
>
> For tasks beyond math/visual reasoning, we conduct an experiment on Referring Expression Comprehension (REC). We train Qwen2.5-VL-3B on 60K data randomly selected from RefCOCO/RefCOCOg/RefCOCO+ using a IoU-based soft reward. The experiment settings are the same as VLM-R1[4]. The results are shown in the table below.
>
> | Model | RefCOCO testA | RefCOCO testB | RefCOCO+ testA | RefCOCO+ testB | RefCOCOg test |
> | --- | --- | --- | --- | --- | --- |
> | Base* | 86.09 | 75.64 | 81.71 | 66.93 | 72.39 |
> | GRPO | 89.90 | 81.33 | 85.94 | 70.97 | 81.45 |
> | Shuffle-R1 | 91.83 | 84.31 | 87.84 | 76.27 | 86.07 |
>
> *: Base model uncontrollably outputs boxes in different formats (absolute, 0-1 relative, or 0-1000 normalized). We transform the detected 0-1 relative boxes to absolute boxes when computing IoU, resulting in lower accuracy on some test set compared to official tech report.
>
> The results above show that Shuffle-R1 also has good performance on 32B model and REC task.
>
>
> ## R.Q2
> We appreciate your suggestion. Additional failure-case and limitation analysis are helpful for deeper understanding of our algorithm and future research.
> In our experiments, we do observed several scenarios where the improvement brought by Shuffle-R1 is less pronounced. As shown in the paper, the gains on HallusionBench over strong baselines are relatively small. We also conducted an additional evaluation on LogicVista, a visual-IQ-style reasoning benchmark with more domain shift, and observed a similar trend:
>
> |Qwen2.5-VL-3B | HallusionBench | LogicVista |
> | --- | --- | --- |
> | Base | 59.83 | 31.69 |
> | GRPO | 63.09 | 35.04 |
> | DAPO | 63.24 | 35.26 |
> | Shuffle-R1 | 63.19 | 34.88 |
>
> We believe this is partly because of significant distribution shifts and harder structured-reasoning patterns that are not fully captured by the reward signals used during RL. Shuffle-R1 is particularly effective when the reward function can reliably reflect preference-consistent reasoning, whereas highly abstract multi-step reasoning (e.g., LogicVista) may require different or finer-grained intermediate supervision.
>
>
> ### References：
>
> [4] VLM-R1: A Stable and Generalizable R1-style Large Vision-Language Model, arXiv:2504.07615

---

> ### Author Response · Authors · 2025-11-26
> **Looking forward to hearing from you**
>
> Dear reviewer `Codo`,
>
> Thank you again for your valuable feedback.
>
> In response to your questions, we have added approximate theoretical analysis on training dynamics, as well as a discussion with representative curriculum-based methods. We have also conducted 32B scale experiment and Referring Expression Comprehension experiment to verify the scalability and generalization capability of Shuffle-R1. In addition, we have provided more design comparison results and a brief failure-case analysis as requested.
>
> Your feedback has been invaluable in refining and strengthening our work. We appreciate your time and effort again.
>
> As we are now actively engaged in the Author–Reviewer Discussion phase, it would be great to have your further feedback at your convenience.
>
> We are looking forward to your reply and further thoughts!
>
> Best regards,
>
> The Authors of Submission 8742

---

> > ### Comment · Reviewer_Codo · 2025-11-28
> >
> > Thank you for your response, which has addressed my concerns. I would encourage the authors to incorporate the relevant discussions into the final manuscript, particularly the theoretical analysis, the comparative analysis with curriculum-based sampling, and the experimental results on scaling to the 32B model size.

---

> > > ### Author Response · Authors · 2025-11-28
> > > **We appreciate your acknowledgement**
> > >
> > > Dear reviewer `Codo`,
> > >
> > > We are happy to hear that you are satisfied with our response. We will continue to refine and polish the paper as you requested. Thank you again for your time and effort!
> > >
> > > Best regards,
> > >
> > > The Authors of Submission 8742

---

### Official Review · Reviewer_5QmF · 2025-11-04

**Soundness:** 3
**Presentation:** 4
**Contribution:** 3
**Rating:** 6
**Confidence:** 4

**Summary:**

The paper proposes Shuffle‑R1, a data-centric RL training wrapper for multimodal LLMs that addresses advantage collapsing and rollout silencing by pairing high-contrast trajectories and shuffling batches based on advantage magnitude. The method is simple, practical, and empirically effective, showing improved accuracy and efficiency over strong RL baselines on math and vision-language benchmarks.

**Strengths:**

The strengths of the paper are:

- Simple, practical method that is easy to implement on top of GRPO, which addresses a real pain point in RL fine-tuning: many trajectories are statistically uninformative.
- Results are strong and it shows effectiveness on the good coverage of in-domain and out-of-domain benchmarks.
- Detailed experiments with ablation studies.

**Weaknesses:**

The weaknesses of the paper are:

- Theoretical analysis of bias/variance under selective sampling is limited; unbiasedness is not proven.
- Missing some clarifications and ablation study

**Questions:**

1. While steps and batch sizes are reported, it is not fully clear that all baselines (GRPO/DAPO/GSPO/RLOO/Reinforce++) were run with identical token budgets, rollout counts (2N=16), and decode temperatures. Small differences can swing math benchmarks materially. Is it possible to add a compute-matched table and a training curve wall-clock plot to substantiate the "7% overhead" claim across settings?

2. HallusionBench results improve, but do PTS/ABS reduce refusal or increase over‑assertion? Any calibration metrics or abstention analysis?

3. How does the pair selection affect solution diversity? Any evidence of collapse in reasoning styles (e.g., fewer distinct CoT patterns)?

4. Are there tasks where advantage distributions are already well‑spread (e.g., dense/step rewards), making PTS/ABS less helpful? Negative results would help practitioners choose whether to use the proposed method.

---

> ### Author Response · Authors · 2025-11-20
> **Response to Reviewer 5QmF (1 of 2)**
>
> Thank you for your feedback. Below is our response to your concerns and questions.
>
> ## R.Q1
> We provide a more detailed list of important parameter setup to better clarify the experimental settings.
>
> | Parameter | Baselines | Shuffle-R1 | Note |
> | --- | --- | --- | --- |
> | Token Budget (train) | `max_prompt_len`=2048, `max_response_len`=2048 | `max_prompt_len`=2048, `max_response_len`=2048 | identical |
> | Token Budget (eval) | `max_prompt_len`=6144, `max_response_len`=2048 | `max_prompt_len`=6144, `max_response_len`=2048 | identical |
> | Rollout Size | N=8 | 2N=16 | Shuffle-R1 use $\alpha=0.5$ to align update size |
> | Decoding Temperature (train) | 1.0 | 1.0 | identical, ensure exploration |
> | Decoding Temperature (eval) | 0.5 | 0.5 | identical |
>
> For compute overhead analysis, we have added a wall-clock train/val accuracy plot in Section 4.2 of the revised paper. As shown in the figure, Shuffle-R1 achieves substantially higher train/val accuracy than the GRPO baseline in the very early training stage. More importantly, the total GPU time of Shuffle-R1 only increases by 4% - 7.7% relative to GRPO. When targeting the same accuracy as GRPO, Shuffle-R1 requires roughly half the number of training steps and approximately 60% of the total wall-clock time.
>
>
>
> ## R.Q2
> We appreciate you pointing out the importance of refusal behavior and calibration. We conducted additional analyses to examine whether Shuffle-R1 affects refusal tendencies or induces over-assertion.
>
> Since the model is required to generate binary yes/no answer, we estimate response confidence through a “thinking–answer consistency” metric assessed by `Gemini-2.0-Flash`, evaluating:
>
> - Internal coherence of the reasoning trace (0–2)
> - Use of visual evidence (0–2)
> - Alignment between reasoning and final answer (0–2)
>
> Scores are then normalized to [0, 1]. We define the below metric: (1) Pseudo-refusal: confidence < 0.3; (2) High-confidence error rate: Incorrect rate of response confidence > 0.7; (3) Calibration: Expected Calibration Error (ECE) with step size 0.1. The results are shown in the table below.
>
> | Metric | Base | GRPO | Shuffle-R1 |
> | --- | --- | --- | --- |
> | Pseudo Refusal Rate ($\tau=0.3$) | 32.81% | 7.05% | 1.26% |
> | High Confidence Error Rate ($\tau=0.7$) | 21.77% | 32.28%  | 28.84% |
> | Expected Calibration Error (ECE) | 0.3121 | 0.3342 | 0.3250 |
>
> Shuffle-R1 substantially reduces refusal rate. The high-confidence error and ECE slightly increase over the base model, but remain comparable to GRPO. This indicates Shuffle-R1 does reduce refusal and increase over‑assertion to some extent.
>
> The observations are consistent with prior findings on RLVR. Researches have shown that RL training typically reduces response entropy, leading to more self-consistent and less hesitant outputs[1,2,3]. This naturally lowers refusal rate and may slightly increase high-confidence errors. Crucially, Shuffle-R1 does not amplify this phenomenon beyond GRPO baseline.
>
>
>
> ## R.Q3
> Thank you for raising the question of solution diversity and potential convergence of reasoning styles. To quantify this, we conducted the following experiment comparing the Base, GRPO, and Shuffle-R1 models.
>
> We randomly sampled 100 evaluation queries and, under identical decoding settings (same prompts/token budget; temperature = 0.5), generated 100 rollouts per query. Each response was embedded using `Qwen3-4B-Embedding` model, and we computed the mean pairwise cosine diversity for each model. Results are shown below:
>
> | Metric | Base | GRPO | Shuffle-R1 |
> | --- | --- | --- | --- |
> | Cosine Diversity | 0.217 | 0.138 | 0.137 |
>
> Both RL-trained models show reduced diversity compared to the Base model, indicating a mild convergence of reasoning patterns. As discussed in **R.Q2**, such entropy reduction is a well-known consequence of policy optimization rather than a failure of our pruning strategy. Importantly, diversity of Shuffle-R1 outputs is comparable to GRPO, suggesting that our algorithm does not introduce additional convergence towards fewer CoT patterns beyond current paradigms.
>
> ### References：
>
> [1] The Entropy Mechanism of Reinforcement Learning for Reasoning Language Models, arXiv:2505.22617
>
> [2] Maximizing Confidence Alone Improves Reasoning, arXiv:2505.22660
>
> [3] Reasoning with Exploration: An Entropy Perspective on Reinforcement Learning for LLMs, arXiv:2506.14758v2

---

> ### Author Response · Authors · 2025-11-20
> **Response to Reviewer 5QmF (2 of 2)**
>
> ## R.Q4
> Investigating the generalization capability of algorithms is valuable. Consistent with RLVR paradigms such as GRPO/DAPO, Shuffle-R1 primarily focuses on outcome reward RL in sparse reward scenarios, while its algorithmic design and code implementation are fully compatible with dense reward settings. To further verify its generalization ability, we conducted experiments on the Referring Expression Comprehension (REC) task, a subset of Visual Grounding that adopts IoU-based soft rewards.
>
> Following the experimental setup of VLM-R1[4], we performed training on the Qwen2.5-VL-3B model using 60k samples randomly sampled from the RefCOCO/RefCOCOg/RefCOCO+ datasets, with a total of 500 training steps. The detailed experimental results are presented below
>
> | Model | RefCOCO testA | RefCOCO testB | RefCOCO+ testA | RefCOCO+ testB | RefCOCOg test |
> | --- | --- | --- | --- | --- | --- |
> | Base* | 86.09 | 75.64 | 81.71 | 66.93 | 72.39 |
> | GRPO | 89.90 | 81.33 | 85.94 | 70.97 | 81.45 |
> | Shuffle-R1 | 91.83 | 84.31 | 87.84 | 76.27 | 86.07 |
>
> *: Base model uncontrollably outputs boxes in different formats (absolute, 0-1 relative, or 0-1000 normalized). We transform the detected 0-1 relative boxes to absolute boxes when computing IoU, resulting in lower accuracy on some test set compared to official tech report.
>
> The results show that Shuffle-R1 also has good generalization ability on the REC task.
>
>
> ## R.Bias analysis
>
> Thank you for your suggestion. We have added a approximate bias analysis in Appendix H.2 of the revised paper. Since Shuffle-R1 introduces adaptive selection & resampling, the expectation of final advantage receives an additional bias.
>
> Assuming the original distribution of Advantage follows Gaussian distribution $A \sim \mathcal{N}(0, \sigma^2)$ (consistent with probe analysis in Fig. 1), we approximate the PTS process using truncation with threshold $\tau>0$, the retained set can be expressed as $S^{\text{PTS}}=\{ |A|>\tau \}$.
>
> The expectation of final advantage is expressed as:
>
> $\mathbb{E}[A] =\frac{1}{Z} (\int_{-\infty}^{-\tau}a|a|p(a)da + \int_{\tau}^{\infty}a|a|p(a)da) = \frac{2}{Z} \int_{\tau}^{\infty}a^2p(a)da > 0$.
>
> where the normalization factor $Z$ is defined as:
>
> $Z = \int_{-\infty}^{\infty}|a|p(a) \mathbb{I}(|a|>\tau)da =2 \int_{\tau}^{\infty}a \cdot p(a)da$
>
> The positive bias indicates that the gradient update systematically tend to trajectories with higher advantage. Under the policy gradient paradigm, it is equivalent to increasing the sample probability of high advantage trajectories, consistent with the idea of algorithms like REINFORCE++.
>
>
> ### References：
>
> [4] VLM-R1: A Stable and Generalizable R1-style Large Vision-Language Model, arXiv:2504.07615

---

> ### Author Response · Authors · 2025-11-26
> **Looking forward to hearing from you**
>
> Dear reviewer `5QmF`,
>
> We'd like to thank you again for your constructive review.
>
> In response to your suggestions, we have provided a more detailed list of important parameters, added a direct comparison plot of wall-clock training time. We have also included deeper analysis on HallusionBench and mode CoT patterns. In addition, we conducted extended experiment on Referring Expression Comprehension task to further investigate the generalization capability of Shuffle-R1. To provide a stronger theoretical support, we have added an approximate analysis of training dynamics and bias to the revised paper.
>
> Your feedback has greatly helped us improve the clarity and completeness of our work.
>
> We are actively participating in the Author–Reviewer Discussion phase and would be happy to provide further clarification if needed.
>
> We would greatly appreciate your further feedback. Looking forward to your thoughts!
>
> Best regards,
>
> The Authors of Submission 8742

---

### Author Response · Authors · 2025-11-20
**General response**

Dear reviewers, ACs and PCs:

We sincerely appreciate your reviews and valuable feedback. We have responded each comment individually in the main response. Below, we summarize the key modifications in the revised paper and provide clarifications for concerns raised by multiple reviewers.

## Modifications of revised paper

We have added a detailed training curve wall-clock plot and discussion in Section 4.2

We have conducted a full setting 30k-data experiment on Qwen2.5-VL-7B to demonstrate the scalability of Shuffle-R1. Results are reported in Appendix C.

We have added additional theoretical analysis of proposed algorithm. An approximated training dynamics analysis and bias analysis in Appendix H.1 and H.2.

The structure of Section 4 and Appendix is adjusted for added content.

## Common Concerns and Questions

Both reviewer 5QmF and Codo request for additional theoretical analysis. We have presented the full analysis process, including propositions and proofs in Appendix H.1 and H.2. We only provide key conclusion in the main response for conciseness.

Reviewer 5QmF, Codo and gi6w request for broader discussion beyond math/visual reasoning tasks or sparse outcome reward. We include a representative study on Referring Expression Comprehension (REC), a subset of Visual Grounding task with IoU-based soft rewards. Following VLM-R1’s setup, we train Qwen2.5-VL-3B on 60k samples randomly selected from RefCOCO/RefCOCOg/RefCOCO+ for 500 steps, and report accuracy on corresponding test sets.

We use the widely adopted prompt for REC: "Please provide the bounding box coordinate of the region this sentence describes: {QUERY} Output the answer in pixel coordinates in the format of [x1, y1, x2, y2]. Minimum value is 0 and maximum value is the width/height of the image. Output the answer in JSON format."

Note: the base model uncontrollably outputs boxes in different formats (absolute, 0-1 relative, or 0-1000 normalized). We convert relative boxes to absolute coordinates during evaluation, and treat normalized bboxes as failure because they cannot be reliably detected. This leads to lower accuracy compared with the Qwen2.5 technical report, and results are not directly comparable to VLM-R1 due to reduced training data.

For open-ended QA and wider tasks/scenarios, they can be either: (1) framed as outcome-reward RL (using rule based reward functions, or LLM judges); or (2) framed as token-wise process reward RL (using reward model). Since application on these topics primarily concern reward function/reward model design, rather than improving current RLVR paradigms, both cases are beyond the scope of the paper. Consequently, we did not include them in the response, and we hope the reviewers would understand.

---

### Meta-Review · Area_Chair_4LUc · 2026-01-08

**Summary:**

This paper proposes Shuffle-R1, a data-centric RL post-training framework targeting two practical inefficiencies: Advantage Collapsing and Rollout Silencing. The method introduces Pairwise Trajectory Sampling (PTS) to select high-contrast advantage pairs and Advantage-based Batch Shuffle (ABS) to dynamically reshape minibatch composition toward informative trajectories. Across multiple multimodal reasoning benchmarks and model scales, Shuffle-R1 improves training efficiency and final performance over strong RL baselines.

Reviewers’ main concerns centered on (i) limited theoretical grounding, (ii) clarity of experimental matching and missing ablations, (iii) comparisons to other data-centric/adaptive-sampling paradigms, (iv) scope/generalization beyond math-like settings and scaling to larger models, and (v) whether improvements were attributable to increased rollout budget rather than the proposed sampling strategy. The authors addressed these points with substantial revisions.

Overall, the revised submission is strong: it offers a practical framework with convincing empirical validation, improved rigor and positioning after rebuttal. I therefore recommend Accept (poster).

**Reviewer Concerns:**

Addressed Concerns:
- Theoretical analysis: Reviewers noted limited formal justification for the proposed techniques. Authors added theoretical analysis/proofs and incorporated clearer explanations.
- Comparison to other data-centric / adaptive-sampling RL methods: Authors added discussion contrasting Shuffle-R1 with curriculum-based approaches and provided comparisons to LIMO/LIMR, strengthening the positioning within the adaptive data-selection literature.
- Benchmark scope / generalization and scalability: Initial experiments were concentrated on math/visual reasoning. Authors added a Referring Expression Comprehension study and a 32B-scale experiment demonstrating scalability.
- Fair comparison of cost vs improvements: Concerns are that gains might come from increased rollout budget or added pipeline complexity. Authors added strict rollout-aligned baselines, wall-clock plots and early-stop comparisons showing faster convergence and improved performance under similar time/compute.

**Reviewer Scores:**

Given that the key concerns raised in the initial reviews were substantively addressed during rebuttal, I expect that each reviewer would either increase their score or maintain their already-positive recommendation.

---

### Decision · Program_Chairs · 2026-01-26

Accept (Poster)